# Effects of the Early Phase of COVID-19 on the Autistic Community in Sweden: A Qualitative Multi-Informant Study Linking to ICF

**DOI:** 10.3390/ijerph19031268

**Published:** 2022-01-24

**Authors:** Anna Fridell, Hjalmar Nobel Norrman, Linda Girke, Sven Bölte

**Affiliations:** 1Center of Neurodevelopmental Disorders (KIND), Centre for Psychiatry Research, Department of Women’s and Children’s Health, Karolinska Institutet & Stockholm Health Care Services, Region Stockholm, 113 30 Stockholm, Sweden; hjalmar.nobel@ki.se (H.N.N.); linda.girke@psyk.uu.se (L.G.); sven.bolte@ki.se (S.B.); 2Child and Adolescent Psychiatry, Stockholm Health Care Services, Region Stockholm,118 61 Stockholm, Sweden; 3Curtin Autism Research Group, Curtin School of Allied Health, Curtin University, Perth, WA 6102, Australia; 4Department of Psychology, Developmental Psychology, Uppsala University, 751 05 Uppsala, Sweden

**Keywords:** COVID-19, autism, ASD, neurodevelopmental disorders, pandemic, ICF, ICF-CY, Sweden

## Abstract

While the COVID-19 pandemic is ongoing, early outcome studies indicate severe and pervasive global effects of the pandemic and associated measures to prevent the spread of the virus. General population studies, as well as insight into the outcomes for particular groups, will be necessary in order to mitigate potentially long-term effects as well as to prepare for future epidemics or pandemics. The pandemic conditions have been marked by rapid and abrupt changes and unpredictability which are circumstances that leave the autistic population particularly vulnerable to adverse outcomes following the distinctive features of the diagnosis. Studies are only beginning to delineate the outcomes of the global autism community and the present study adds to these findings by providing a local, multi-perspective, qualitative analysis of the lived experiences of the Swedish autism community. In this study, autistic youth and adults, caregivers of autistic individuals, as well as representatives of Swedish interest organizations were interviewed. Thematic analysis was performed on the population as a whole and patterns of results were formalized according to the International Classification of Function, Disability and Health (ICF-CY). Participants report wide-ranging adverse outcomes of the pandemic relating to mental health and access to support, participation in daily activities and socialization, education, and work as well as parental resources. However, participants also report positive outcomes relating to a reduction in specific social and everyday demands, and normalization of lived experiences. Additionally, interviews outlined some strategies used to cope during pandemic conditions. Implications of these findings are discussed.

## 1. Introduction

The COVID-19 pandemic has disrupted many areas of people’s lives—including work, education, leisure, socializing, various forms of support and healthcare, and everyday duties. Global effects of prevention measures—broadly including abrupt closure of essential services, social distancing, and enhanced hygienic precautions—aimed at reducing the spread of the virus are unfolding [1]. While knowledge of outcomes on the general population is essentially moving forward, insight into the impact of COVID-19 on specific populations will further enhance the potential to prepare for future epidemics and pandemics as well as provide targeted intervention and support measures for particularly affected groups. Additionally, considering the variability of how prevention measures have been implemented globally [2], local analyses of vulnerable populations may elucidate specific strategies that may mitigate some of the risks of long-term adverse effects. Autism spectrum disorder (henceforth autism) is a neurodivergent condition defined by challenges in social interaction and communication along with repetitive, restricted behaviors and interests as well sensory processing alteration causing varying functional impairment and need of support [3,4,5,6].

The distinctive diagnostic features and functional impairments are associated with a preference for highly predictable environments [3]. Conversely, change, unpredictability, and uncertainty are circumstances under which autistic individuals may respond with significant emotional and behavioral distress [7,8,9,10,11,12]. Moreover, caregivers of autistic children experience higher levels of stress than other groups of parents during pre-pandemic conditions [13] relating to the lifelong needs of their child, parents’ pivotal role in advocacy, care coordination and provision, as well as typically variable access to targeted formal support [14]. Therefore, it is expected that the unpredictability, rapid and frequent changes, and transitions that have marked the crisis mode during the pandemic in terms of e.g., educational or occupational status, daily life routines, hygienic procedures, and treatment protocols are particularly damning for this group. Additionally, more time spent at home and changes or reduced availability of support services across life areas likely put further strain on coping during the pandemic for autistic individuals and their caregivers [15].

Indeed, international studies are accumulating to support these fears. Parents of children with autism report wide-ranging adverse outcomes including increasing sleep issues [16], less physical activity [17], disruptive and restrictive behavior, and signs of regression in younger children [18], higher levels of irritability, behavioral problems, stress, and anxiety along with less adaptive coping strategies in school-aged children [19,20]—especially for children with pre-pandemic behavioral and psychological issues [21]. 

Clearly, the pandemic conditions have had pervasive effects on the lived experiences of the global autism community. Notably, several studies are now showing that families have at least partly lost access to healthcare and support services [22,23,24,25] causing worries about the potential long-term effects on children’s emotional health as well as educational and developmental progress [23]. Nevertheless, adhering to prevention measures and guidelines, many care services changed the structure or adapted the content of treatment protocols to move online [24,25]. Parents have reported variable satisfaction rates with remote services, being generally satisfied with remote psychotherapeutic services but dissatisfied with behavioral, speech/language, and occupational therapeutic interventions remotely. Similarly, many educational institutions have transitioned to remote teaching, adapted face-to-face teaching, or a hybrid version of the two, resulting in changes to educational and special educational support practices for autistic students [26]. While efforts to maintain educational practices have been commendable in many respects, studies report variable outcomes for autistic students. Parents report disrupted transition planning in younger children [18] and report that schools have not been adequately considering the needs of their child or modified their expectations of the child during pandemic conditions [22]. 

The adult autistic community report similar adverse outcomes including increased anxiety, depression, loss of social support systems, worries about losing social skills with less social interaction, loss of regular care services, and added communication challenges with remote service provision [27,28]. Additionally, Taylor et al. [29] found that a large portion of autistic adults has changed employment status (e.g., loss of job, reduced hours) during the pandemic which in turn was associated with depressive symptoms.

Additionally, studies are beginning to elucidate how these outcomes are affecting parental resources. Comparative, population-based survey data show greater psychological distress (including symptoms of anxiety, loneliness, and hyperarousal) in parents of children with in comparison to parents of children without autism during the pandemic [30]. Summarizing outcome studies, Yalmaz et al. [31] describe that parents’ mental health is related to the child’s ability to adapt to changing routines, dealing with social distancing and enhanced hygiene practices, and serious behavioral disruptive behaviors but also economic difficulties.

Conversely, while adverse outcomes during COVID-19 in the autism community are pronounced and require careful consideration, studies report some better-than-expected outcomes. In Adam-Mumbardó et al. [32], parents report that most children appeared happier as well as communicated and participated in family routines more during quarantine. Further, families appreciated the increased workplace flexibility and appear to have benefited from the extra time shared at home—e.g., by training skills and improving family dynamics [18,23,28]. In addition, Marín-Lugo et al. [33] assessed psychopathological symptoms and life conditions changes following lockdown using retrospective medical records and online surveys with parents and autistic adults. Although the study reported an increase in caregiver stress, findings suggest a general decrease in psychopathological symptoms in autistic youths and adults and a reduction in self-perceived stress in autistic adults. In addition to treating adversity, intervention planning may act by strengthening factors that support well-being or healthy adaptation following adversity, i.e., protective factors for resilience [34,35], to mitigate the risks of long-term negative outcomes in the autism community following the pandemic and may therefore be considered in more detail in further studies. 

Overall, however, less is known of the outcomes following the COVID-19 pandemic on the Swedish autism community. Compared to more comprehensive actions taken to manage the pandemic elsewhere in comparable Scandinavian countries [2], the strategy chosen with the Swedish authorities included less enforced restrictions to public life. Nevertheless, restrictions in Sweden did include, among other things, remote teaching for high schools and universities, physical distancing from close ones, a shutdown of cultural and sports events, working from home, as well as care services adhering to physical distancing whenever possible (see Appendix A for information about actions in Sweden at the time on this study). Thus, many in the autism community will have experienced changes in routines and opportunities in several life areas, challenges to typical forms of communication with important others, and overall increasing unpredictability wherefore stress and worry are likely to increase. To the authors’ knowledge, two studies have been published on the outcomes in the Swedish autism community—both relating to education (teacher perspectives [36]; cross-country parental survey [37]. Thorell et al. [37] specifically indicate Swedish students had comparably more contact online with teachers (24.38%) and other peers (12.87%) as well as less schooling with parents (15.28%) in relation to other European countries. Notably, Swedish parents report quantitatively lower levels of negative effects of homeschooling on parents (11.1%), less parental worry and stress. Moreover, more positive than negative effects for the child are reported, but more negative than positive effects for parents—in accordance with caregiver outcomes in other studies.

This study aimed to explore the lived experiences and impact of COVID-19 on the Swedish autism community through semi-structured interviews with autistic youths and adults, as well as parents of autistics and representatives of autistic interest organizations in Sweden. Multi-informant qualitative results add to previous international studies primarily using online parental survey data (e.g., [18,37]). The present study inventories the negative and positive outcomes on the lived experience in the Swedish autism community in Sweden during the initial phase of COVID-19. For formalizing the pattern of results from the qualitative study, interview information was also linked to the bio-psycho-social framework and common language of the International Classification of Functioning [38] and the ICF core sets for autism [4].

## 2. Materials and Methods

### 2.1. Sample

In total, 38 individuals participated: (i) 7 autistic children/youths aged 14–17 years (M = 16), 3 identified as girls, (ii) 13 autistic adults aged 18 or above (M = 33), 10 identified as women, (iii) 13 parents (age range 38–53 years) of autistic children (age range 6–17 years), and (iv) 5 representatives of interest organizations (age range 30–70; 2 women). Schooling status of autistic children/youths was collected through both demographic questionnaires and transcripts of interviews. The majority of participants experienced no change in occupational status, one autistic adult changed from part-time to full-time occupation following the start of the pandemic. Several areas of Sweden are represented in the samples, albeit with a majority residing in or around the capital Stockholm. Please refer to Table 1 for further details on the population of autistic children/youths, adults and parents regarding experiences with COVID-19, co-occurring conditions, household occupants, occupational or educational status, as well as the self-reported degree of adherence to COVID-19-related restrictions. 

For an overview of the characteristics of the subjects included in the focusgroup interview of representatives of autism interest orgaisantions, please refer to Table 2.

A priori, two participants (both parents) who had consented were excluded, as they did not follow through with interviews for personal reasons. Participants were recruited through purposive sampling—aiming for distribution between the person-categories—by advertisement in print and social media channels related to our lab—and major Swedish interest organizations for people with neurodevelopmental conditions. Sufficient Swedish language proficiency was required as all interviewees needed to complete interviews in Swedish only. Autism diagnosis was established by self-reported clinical diagnosis according to regional health care guidelines. Participants were compensated with a gift card worth SEK 250 (≈EUR 25).

In one interview, two parents participated in the same interview—these were analyzed as a single parent experience. For five of the autistic youths, we also interviewed one or both of their parents. These dyads are classified and analyzed as separate individuals in the present study. Six of the parents who participated had one or more children with autism who were not interviewed themselves. During interviews, parents were explicitly asked to focus on the experience of one of their autistic children. Two of the parents who participated had themselves a diagnosis of autism. These parents were classified as parents in the demographic overview. For analysis, however, experiences relating to their “parenthood” and experiences relating to “autistic adulthood” were analyzed separately.

### 2.2. Procedure

The study follows a qualitative cross-sectional design and has been approved by the Ethical review board in Sweden (DNR: 2020-03053). Participants who expressed interest were contacted by a member of the research team by telephone and informed of the study and provided age-adapted written, informed consent by digital signature. For participating children aged 12 to 15, both parents and the child were required to provide written consent for participation. Following consent, participants were invited to complete a short (10 to 15 min) age- and group-specific online questionnaire on demographic and clinical information, such as age, gender, other diagnoses and health issues, education level, and occupational status collected through the Karolinska Institutet Survey platform. For parents, this included both questions about themselves and their child. Autistic children aged 12 to 14 years were given a targeted set of questions in the interview (two participants) whereas autistic children aged 15 to 17 years completed adult-oriented questionnaires and interviews. Representatives of interest organizations responded to questionnaires regarding general demographic information as well as active years and role in interest organizations.

Semi-structured interviews were conducted between June and August of 2020 (see Appendix A for details regarding the Swedish response to the pandemic and local circumstances during this time) by members of the research team with clinical experience of the target population as well as research protocols. The majority of interviews were conducted over a secure videoconference system (Zoom via Sunet); two participants (both young persons) preferred to respond in written form.

### 2.3. Interview

The main interview topics and example questions were tailored to age-specific groups and representatives of interest organizations and provided to participants beforehand. These covered the impact of the pandemic on (1) family and housing, (2) education and employment, (3) mental and somatic health, (4) support and service provision, (5) social relationships, (6) personal interests and activities, (7) outlook on the future, and (8) potential positive experiences of the pandemic. Each topic was introduced by asking one or several open-ended questions, followed by open or closed follow-up questions as the discussion progressed. Interview lengths with autistic participants and caregivers ranged between 27 min and 35 s and 78 min and 11 s (median = about 48 min) and interview length with representatives of interest organizations was about 1 h and 3 min. Time spent on each topic varied between participants, depending on their individual situation, concerns, and interests. Interviewers actively inquired about any side-effects following the interview and, upon indication, participants were advised and referred to support services according to the study protocol.

### 2.4. Analysis

#### 2.4.1. Thematic Analysis

All of the interviews were transcribed verbatim by an external professional transcription service. Transcripts with autistic participants and caregivers ranged between 3366 and 10,012 words (written interviews M = 1461) and the transcript with representatives of interest organizations consisted of 8844 words. Transcripts were pseudonymized and given an identification number by the research team. Aiming to report the experienced reality of study participants a reflexive thematic analysis was conducted as guided by Braun and Clarke [39,40] using version 12 of NVivo software. Analysis was done with a realist/essentialist framework (i.e., with the focus on deriving meaning from the participant’s language) while focusing on the semantic level of the dataset [39]. Further, data analysis was done inductively initially, whereby patterns of meaning guided coding rather than any pre-existing coding schedule.

Throughout the data collection and transcription phases, members of the research team regularly met to share initial reflections and take notice of specific observations in order to contribute to analyses. As transcription was finalized, one member of the research team (A.F.) was further immersed in the data set by reading and re-reading the transcripts while taking notes of salient features and coding transcripts. Codes were generated for the full dataset including autistic children and adolescents, autistic adults, and parents. While certain lived experiences were naturally specific to certain life areas (e.g., the main theme “meeting the challenge of remote learning”) and therefore excluded participants not included in the specified context (e.g., non-studying participants), codes and themes were created to reflect commonalities between all or most participants. Throughout the analysis, codes were generated for autistic experiences exclusively. Undiagnosed parental experiences were included when they related to their autistic child. Parent–child dyads we included in the analyses but analyzed separately as they might complement each other. Discussion between authors informed coding by considering breaking down codes, inserting new codes, and merging some existing codes. Codes with infrequent references or that provided no further or salient information to themes were. Six interview transcripts (two young person, two by adult person, and two parent interviews) were re-coded by another author according to the generated codes which confirmed coding satiety. Codes were categorized into thematic areas. Within these areas positively and negatively valued codes were clustered into sub-themes according to positively and negatively valued outcomes. Qualitative analysis was performed separately for representatives of interest organizations using an equivalent approach to reflect the secondary-level experience expressed by this group, however, themes were not clustered into positively and negatively valued outcomes. Results aimed to further nuance or complement the findings of first-hand experiences of families and are presented separately. 

Swedish to English translations were conducted following analyses for international readership. Translations aimed to enhance conceptual equivalence wherefore some clarifications and alterations have been made to this end [41,42]. Finally, the overall draft analysis was reviewed by a senior member of the research team (S.B.) for refinement.

#### 2.4.2. ICF Linking

ICF-linking was completed for interviews with autistic children/youth, adults and parents exclusively. Following qualitative analysis, thematic codes were deductively linked to the International Classification of Function, Disability and Health (ICF), developed by the WHO as a general framework for describing function and dysfunction [38]. First, we re-read all the statements belonging to a given thematic code. Second, we discussed the components of the thematic code as a whole. Third, we selected the most precise ICF codes corresponding to these components from the ICF: Children and Youth Version (ICF-CY). This version of the ICF entails all categories of the ICF as well as additional categories specifically relevant for developing individuals [38]. Fourth, we cross-referenced these against the ICF core sets for autism [4], which consists of a subset of the most relevant ICF second-level codes for describing functioning in autism. Finally, we report only the specific codes that also appear in the autism core set, but also report the total number of codes initially selected from the ICF-CY. The ICF linking followed established linking rules from the ICF Research Branch [43]. Specifically, components within each thematic code were linked to the most precise ICF code (ICF second or third level) and, when needed, converted to second-level codes to allow comparisons with the ICF core set, which consists of second-level codes. Finally, a senior member of the research team (SB) reviewed the overall draft linking analysis.

## 3. Results

### 3.1. Thematic Analysis of Interviews with Autistic Childre/Youth, Adults and Parents

The thematic analysis yielded 11 main categories; each with negatively and positively valued subthemes/codes. Altogether, themes encompass 61 codes. The main categories as well as negatively and positively valued themes are described below in that order. Quotes are identified by YP (child/adolescent), AP (adult), and PP (parent) and an identification number. Thematic area and sub-themes are denoted in headings. Codes are described in cursive in text and included raw data in quotation marks. See details in Table 3, including frequencies of reporters per code and linked ICF categories.

### 3.2. Health during COVID-19

Circumstances during COVID-19 have had an impact on the psychological and physical well-being of participants. While many expressed detriments to health, some suggested that well-being has been supported.

#### 3.2.1. Detriments to Overall Health

Many participants reported *understimulation to cause distress* during COVID-19—“a sense of meaninglessness” [YP-78] and lack of motivation in everyday routines. Additionally, *restrictions have taken a toll on mental health* as participants reported increased anxiety, lowered mood, and occasional self-harm. For instance, one autistic adult expressed “I usually have a lot of worries and anxiety, but I think it has become worse” [AP-01]. Many also described *continuous health concerns* regarding own and others’ health during the pandemic with one parent expressed their child said “And he was worried, will you die mummy?” [PP-70]. Some also reported challenges including *continuous financial concerns*, such that “But I definitely have financial worries all the time. Because of COVID-19, amongst other things” [AP-06]. In addition, physical health behavior has been affected, including *less daily activity*—as one autistic adult said, “But I have been outside a dozen times at most since March. That might not be super healthy” [AP-06].

#### 3.2.2. Supported Well-Being

There were indications that some had been *thriving during the pandemic*, at least in parts of their lives. Some reported having had more energy to socialize with peers, as there were fewer demands in other life areas during the pandemic, one autistic youth explained, “I have a lot more energy left to do things I like during the week. And that’s because I kind of don’t need to be in the classroom where there can be a lot of noise that can disturb me” [YP-08]. In addition, participants shared that they have had the opportunity to explore new or *spend more time on interests* they are able to do online or by themselves. For instance, one autistic adult said, “But that you realize, damn I can do more than I thought. That you have more time to actually immerse yourself in something and get better at it” [AP-06]. Additionally, some had been able to maintain some physical activity—some *found other ways to stay active*—one adult noted “But then I felt that—no this is going to last for a while so then I will leave. Because I can live more actively here, and it is easier for me to follow restrictions and stuff like that” [AP-32]”, and others reflected that *their pet takes them outside*. 

### 3.3. Access to Support

Participants expressed experiences of how formal (e.g., caregiver services, unemployment services, daily support services) and informal (e.g., elderly relatives) support services in person and online have been available for them over the pandemic. 

#### 3.3.1. Restricted Access to Support

Several participants reported that *finding adequate support is always difficult for us*, and one adult described “I don’t really think I got that support before COVID either (laughter). I have mostly been shuffled around different people or have been told me ‘no’ cause I have autism and they don’t deal with that, or they don’t know anything about that” [AP-31].

Participants reported that *regular support services had become unavailable*—for reasons beyond support services closing down or being canceled during COVID-19. Some explained that they “have been unsure of how things work” [AP-05] or worried about “travelling by public transport or sitting at a healthcare service space with other people” [AP-12] and talking online while their “partner was working from home” [AP-12] wherefore they had opted out from contact with support services. Others had not been able to access unemployment services as their case handlers were overwhelmed or had not been able to access food home delivery services as they are used by more customers—one adult explained that “from March, April and the May it was three and a half week, the delivery times where prolonged. And that didn’t work for me. Because I don’t have the ability to look that far ahead” [AP-35]. Additionally, parents that were usually supported by elderly relatives had not accessed this support due to strict recommendations for this population. Additionally, while available for some, *remote services offer communicative challenges* for some as “moods” [AP-32] are difficult to sense and turn-taking can be harder [AP-32]—as explained by one adult.

#### 3.3.2. Access to Regular Support

Some described that remote services facilitated participation as *remote support allow flexibility*, as one youth described, it made it “easier to talk” [YP-76] and removed practical demands, as one adult put it “if it’s over the phone I can sleep a bit longer and stuff. So that feels much better” [AP-33]. Participants also described that *in-person support has been available as usual*, including short-stay support family, at-home daily life support services, and healthcare services, and even “has been especially important” [AP-14] during the pandemic.

### 3.4. Need for Socializing

Overall, participants described the importance of socialization and the ways they have experienced how this need had been met or unfulfilled over the pandemic. Participants also described the role online socialization has had in this regard.

#### 3.4.1. Sense of Loss of Social Connection

Several participants described *less in-person socializing*—either in frequency or by restrictions of socializing to a select few, as one parent noted “[he] thinks it’s sad that he cannot meet the friends that aren’t in the same pre-school. In some way we have drawn a line there as parents” [PP-70]. Further, many *missed in-person connection with family and relatives* such as elderly relatives and family members that live elsewhere, one youth described “I have reacted quite negatively to that I think, because my grand-father died so I would have wanted to meet him before and kind of comfort my grand-mother” [YP-08]. Additionally, many *missed incidental contacts in regular structured environments* such as in school as students may “usually sit at the same table every lunch” [YP-11] or at work where “there are very clear rules there and everybody talked to each other” [AP-65]. It is reported that *social recreational activities have stopped.* These included individual activities outdoors as well as organized activities—typically associated with social distancing challenges. For instance, one adult described “I had, I really don’t have any close friend so but I have many acquaintances in Pokémon-Go that I meet a lot when I’m out playing, and I have barely seen them at all now” [AP-31]. Alternatives to in-person socializing in recreational activities or with friends online or by telephone had been possible for many but at times *remote connections fall short*—both because “it’s a bit harder to get feedback” [YP-78] as recreational activities move online but there is a sense that remote socializing “is like drinking Cola Zero instead of regular Coke when you are a sugar addict” [AP-65]. Related to the above, several participants reported *feeling lonely* where some young people expressed they “have my parents but they aren’t a substitute for friendships” [YP-73] and that loneliness follows when “social activities have disappeared” [AP-33].

#### 3.4.2. Socializing on My Own Terms

Participants had also been able to continue *in-person socializing with family and relatives* according to recommendations. For instance, a parent reported that they “started a little game, garden café” [PP-70]. Other participants have judged that in-person socializing has been necessary, for instance, “[mother] came and saw me anyway because it kind of was needed because sometimes I feel so bad I can’t be on my own” [AP-33]. Participants also reported that there is a general sense that *normalization of social distancing* where one parent described “We are already experts on this social distancing. So we haven’t had any problems with that part” [AP-12]. Relatedly, several participants described *fewer demanding social contexts* and one young person described “I don’t want any close contact with people. I don’t like talking to other people unless I have a reason to” [AP-35]. Additionally, some participants expressed that *online socializing can fill my social needs*—either by structuring online interaction around watching a movie or playing a game or getting a chance to control interaction according to individualized needs. One youth stated, “It’s been pretty nice that you can finish when you get tired. You don’t have to wait until the other person goes home or anything like that” [YP-78].

### 3.5. Participation in Society

COVID-19 affected participants’ possibilities to participate in various societal activities, including organized recreational activities performed in groups.

#### 3.5.1. Restricted Participation

Some reported that *when others implement restrictions my participation becomes limited* and some reported being worried about being outdoors or public spaces suddenly becoming too crowded. For instance, one parent reported that socializing outdoors was challenging as “he absolutely cannot be outside if it’s wind” [PP-02] and another described that participation in an organized activity was difficult as a result of unpredictable circumstances “…then they were going to be outside, then inside, then it was going to be on a Wednesday, yeah” [PP-07]. 

#### 3.5.2. Participation Was Made Possible

A few reported, however, that *in-person continued participation in recreational activities* continued similarly to before the pandemic, and one adult said that “I feel like it’s necessary for my health in some way. Cause I think I would almost go crazy if I couldn’t” [AP-01]. Moreover, some expressed that *online alternatives can allow participation in recreational activities* such that “Yeah, we had some online-practices that were easiest to attend; it wasn’t like you had to get ready for like an hour beforehand” [YP-08].

### 3.6. Establishing Circumstances When Moving Online

Aspects of the transition to remote working and learning were experienced as both favorable and challenging for participants.

#### 3.6.1. Negative Experiences

Some participants reported that the transition was marked by a *lack of understanding and adequate support.* Changes were occasionally experienced as sudden and one adult noted that “I think we got the e-mail on the evening of the Wednesday, that it [assistance participating in daily activities] would be closing down from Monday…” [AP-04]. A parent expressed some frustration “Because suddenly when all the typical kids or whatever you say, that have no diagnoses, when they suddenly were at home, support was somehow arranged for how you can handle it when kids aren’t really coming to school” [PP-67].

#### 3.6.2. Positive Relationships and Communication

For some participants, circumstances were marked by *trust in staff*, expressed by one young person “So I’ve had good teachers and stuff. But I’ve still had a hard time. And they have understood that” [YP-73]. Others expressed that they *felt prepared as they transitioned* by their employer “There were clear routines for how to do it too pretty early. There were weekly updates from the company board” [YP-73] or from the school they attended “yeah, general understanding… They give us information in time, they, I know they think a lot about that” [YP-78]. Another facilitating circumstance was lack of change, one adult explained, “So, well the usual then, since it was remote-learning from the beginning it, well, continued to be remote in all sort of courses and high school and then stuff started to do remote learning. So, there wasn’t any changes and stuff” [AP-04].

### 3.7. Learning and Working on Site

As for Swedish society as a whole, some participants continued working and schooling on site. These circumstances primarily included pre-school and elementary-aged students as well as adults employed in essential services, including staff at elementary schools that remained open. Nevertheless, restrictions have still affected their everyday living experiences. 

#### 3.7.1. Learning and Working Became Harder

Some participants stated that *in-person routine changes fail to attend to individual needs*. What might be perceived as smaller everyday adaptations disappeared—with one parent saying “we have to leave and pick him up outside of the pre-school all of a sudden. That has been a really hard change”. Moreover, one adult explained, “Those that have come back, including me, had to work double after. So no, I can’t say that. Nothing positive, the opposite. More stress I would say”. 

#### 3.7.2. Learning and Working Was Facilitated

Even other participants felt they *did more of what I’m good at* as a result of pandemic-related changes, one adult said “it is part of my autism-superpower kind of that I am skilled at administrative work. So, I’ve gotten more hours of that” [AP-06]. A few participants also felt that *in-person routine changes target individual needs* for instance in the school environment where “those that were kind of sick got to sit and like wait on Google Meet. And there’s a teacher that show what they were supposed to do. In school” [YP-76]. Moreover, one parent noted that “And I think that, that the group of children has gotten smaller [due to illness at kindergarten] has made things easier for him socially and not made him so tired and exhausted” [PP-02].

### 3.8. Meeting the Challenge of Remote Learning and Working

For student participants at school and university, many have experienced remote teaching and learning during the pandemic. This includes primary school and high school as well as university students. While some aspects were described as helpful, many others have been challenging. 

#### 3.8.1. Specific Barriers of Remote Learning

Participants described specifically that *remote learning prevented targeted support*—one parent noted, “They haven’t been able to give, provide adequate support either to be able to structure the school day for [child’s name]. And been able to have the kind of follow-up that might have been needed to push in the right direction on the right stuff” [PP-09-10]. Moreover, *remote learning and working challenge communication*, one young person noted, “It’s a bit harder to get feedback I guess. The teacher can’t see your face so it gets harder for them to interpret what you want or feel and things like that. And many, especially at my school, and me, have some difficulties asking for help and formulating what you need help with” [YP-78]. Many participants also reported that *remote learning challenge self-management*. One adult is noted that “When COVID-19 hit everything was upended. All physical meetings were canceled. Now everything had to be managed online and I was overwhelmed with course material/online-seminars and similar” [AP-14] and one youth expressed that “It’s been harder to motivate myself for school because the boundary between spare time and school has been a bit erased” [YP-08]. Some participants expressed that they *cannot follow their study plan* during the pandemic—one youth stated that “It hasn’t gone so great, I’ve had to remove several subjects but now I will re-do first year at high school because I wasn’t able to finish what I had to” [YP-11]. Additionally, participants experienced *distractions while working and studying from home*, one young person said, “my siblings are running around and disturb me. They run around and shout and play”. 

#### 3.8.2. Specific Facilitators of Remote Learning and Working

On the other hand, participants describe factors that facilitated remote learning, such as *regular check-ins*, one youth explained that “the teachers have been, but always, very often surveys to see how we’re doing” [YP-73] and one parent expressed that “Information flows, clear and clear structure. Generally, they have handled it very well from school” [PP-77]. Some participants reported *studying was more efficient from home*. For instance, one young person said “It’s not as effortful, so it’s easier to ask the teachers questions” [YP-08].

### 3.9. Parental Resources during COVID-19

#### 3.9.1. Added Stressors

Parents of autistic individuals reported that stressors included *communicative challenges between school and parents*. It is noted “What we’ve had to do is to is to try to have communication with the pedagogues at, the special education pedagogues, teachers and principals and stuff at school to an extent. But they have had very little capacity too” [PP-09-10]. Moreover, several parents felt increased stress having *added pedagogical responsibilities*—primarily, since their school-aged child was learning remotely, one parent said “it’s been ok over that period but it has been full focus on that, it hasn’t been like I’ve been able to work during that time, those days, or study” [PP-03]. Another stressor revolved around parents having to put extra effort to *remind and deal with new routines for their children during the pandemic*. For instance, due to changes in regular routines parents had to activate children on top of work-related activities; “Right now I sit for an hour. But then usually my kids get bored and then it’s a struggle anyway. But I just try to manage. But it’s hard. I’ve been on sick leave half time for a period […] I have an anxiety disorder that has become much harder because of COVID” [PP-67].

#### 3.9.2. Reduced Stressors

A few parents expressed that the pandemic has entailed some aspects of stress reduction in the form of *fewer everyday routines*, which can allow for more face-to-face time. One parent expressed that “the mornings used to consist of 20 activities and now they consist of three. It’s amazing. I hope that may stick around even after COVID” [PP-67].

### 3.10. Thoughts about the Future

Overall, participants described some anxiety about their circumstances looking ahead—either if the pandemic would continue for a while or following the pandemic. Nevertheless, several participants described optimistic thoughts regarding lessons for the future and the care members of society might show each other in the future.

#### 3.10.1. Anxiety

Participants expressed general concern in that they are *worried about the future of my society* “it’s an enormous economic hit to society. And we already see the tendencies how you save a lot particularly on people with neuropsychiatric disorders unfortunately” [PP-13]. Crucially, several participants expressed that they were *worried about my future* including going back to school or higher education—both because of potential difficulties adhering to prevention measures but also returning to regular school routines. For instance, one youth expressed that “I worry a bit about if I will feel worse and if it will be a bigger shock to return to school” [YP-08]. Other participants expressed worries about their support system, employment, and financial situation going forward “And it’s like, if the world economy does not work then my economy won’t work that well either. So that’s probably what I’m most worried about” [AP-33]. One parent also expressed concern regarding their child that was “on the way to adulthood, that is on the boundary now between finishing her studies in a few years and go into adulthood. Where it’s been clear that it’s partly delayed because of COVID-19 when she has to re-do a year” [PP-09-10].

#### 3.10.2. Optimism

Nevertheless, several participants were optimistically looking ahead and wished for learned *lessons in society in general*—about relations and being social on your own terms, practical lessons and how to deal with crises in the future as well as socio-political lessons regarding, amongst other things, the environment. One adult expressed “After Corona…Well, to be able to hug again, I hope” [AP-01]. Some participants expressed learned *lessons about myself* “But I realize too how stressed I become from demands when I can’t really del with them. That’s been a pretty good insight” [AP-12]. Specifically, also, participants hoped for *digital options* for adaptable access to work, education, and recreation. For instance, one adult said “have proved that this works. And that you hopefully can keep some of these elements when we go back to some kind of normal again. That you maybe can have that a few times, that this I can do digitally” [AP-06]. Others reflected on *developed school platforms* and wished schools would be able to learn from the new experiences and challenges to provide targeted support and that knowledge about the importance of the school environment will be enlightened—one parent explained “The only thing I guess I am happy about is as a teacher that it has been highlighted how important school and education is in different ways. I guess that’s it, as I said, as I mentioned previously what function school fills that have different functions” [PP-03].

### 3.11. Daily Life at Home

#### 3.11.1. Increasing Pressure

Some report they are *generally dissatisfied with routines* “My sleeping has always been pretty bad and it still is” [AP-12]. During the pandemic, participants expressed that *when you lose outside structure days become more challenging/stressful* including structure given by school-organized recreational activities as well as access to public spaces. For instance, regular everyday routines may be harder to keep up given the loss of day-to-day activities—one youth said “But structure from the outside and structure that I create myself. And when you lose what is outside it becomes a bit hard” [YP-78]. Loss of everyday structure challenged some parents’ reliance on familiar excursions “You made up little chores. Like household- or garden work that he had to solve. That way he kept busy” [PP-13]. In addition, participants expressed that the pandemic affected relationships or aspects of relationships. Some reported *strained familial relationships* ranging from weariness and irritation from lack of privacy or time alone, to severe relational strain; for instance, while being forced to relate to each other in a new way in a closed space or disagreement regarding interpretation of restrictions. For instance, one adult described that “A ‘regular’ living situation throughout the spring. Then it was really over the last week we decided to get a divorce” [AP-65] and one parent expressed, “Then communication gets very limited, and he gets very squared. Then I raised my hand and he spinning metal, this medal, metal towards me and my finger broke” [PP-13].

#### 3.11.2. Release and Support

Conversely, while some reported that they are *generally satisfied with routines* such that “So we have pretty established routines. And we largely hold on to them because it usually turns out for the best” [PP-07], some reported a *release of everyday demands* where the pressures of day-to-day tasks, e.g., getting ready to leave and using public transport for school, and environments, e.g., school being noisy, have decreased with beneficial consequences such as sleeping in and starting having lunches every day. For instance, one adult described that “I think stuff like my sleep that it’s worked better because my schedule has been marginally reduced and I can do it” [AP-06] and one parent explained “since she doesn’t get all that, impressions at school, and have to socialize with lots of group-work and talk to people and sounds everywhere. Then she’s been much more alert, then she has energy to socialize with others” [PP-07]. Additionally, some reported *stronger family connections* as the family was spending more time at home resulting in more opportunities for face-to-face interaction over the day; one youth expressed that “Yeah. It’s, really it’s, we have been able to see each other more. Since he [brother] is not with his friends at the moment but he’s here. So, it’s pretty cool and fun that he is here” [YP-72], and a parent that reported that “Within the family it’s actually only been positive for us. It has gotten us more attached than detached. It’s the opposite, we have gotten it much better” [PP-75].

### 3.12. Thoughts about COVID-19 Restrictions

From the beginning of the pandemic, Sweden’s strategy has been to minimize the people being sick and needing care to decrease pressure on healthcare services. As opposed to more aggressive lockdowns, recommendations and guidelines have been communicated and have varied over time with new information. Participants are reporting variable reception of these circumstances.

#### 3.12.1. Distress

For some participants, *understanding and implementing restrictions cause distress* since it could be difficult to handle the responsibility of or to remember implementing prevention measures in their own life. Some wished that various actors had been more flexible with the implementation of restrictions for organized activities or ensured implementation to a greater extent in public spaces. For instance, one parent wished for “somebody to hold my hand and say you’re doing it right, this is how you’re supposed to do it or I don’t know. I think it’s difficult. Not be bothered that everyone is doing things differently all the time” [PP-70], and another parent that “Everyone can do aerobics but then maybe then you consider that perhaps it’s this particular group that would need it most of all and could continue with the activity” [PP-13]. Other than that, several participants explained that while they may have been able to both understand and apply prevention measures in their own life, *they get upset when other people do not comply with restrictions*. One parent explained, “He’s been incredible frustrated by the way the school has not followed the Public Health Agency’s guidelines and stuff. It has really affected his every day to a large extent” [PP-71]. 

#### 3.12.2. Understanding

It was also reported by participants that *more aggressive measures would have had a worse effect*, that they appreciate being in Sweden of all places at the time of COVID-19; one parent noted “No I think it’s been great to have a choice. I don’t think we would have been able to manage that kind of total limitation there has been in other places” [PP-02]. Additionally, some participants reported that they *already observe physical distancing* wherefore restrictions have meant little change in their day-to-day for various reasons; one adult noted, “I have been on sick leave over soon to be two years, for depression. And then I was already, or because of that and already before I have been more or less self-isolated” [AP-35] and one parent said “We don’t have kids with many recreational activities and we don’t kind of socialize every weekend with other families with kids and we don’t travel and run around like that. Where there are actually restrictions now” [PP-09-10]. Relatedly, some participants also reported an *acceptance of restrictions* by finding them manageable and being able to translate them to their own circumstances (either by themselves or through their parents). For instance, one adult noted “Well it’s not positive but you still understood why and that it’s still, well you know, well hopefully that you know it’s for a limited period kind of like” [AP-04].

## 4. ICF Linking Results

ICF linking of interviews with members of the autism community (i.e., autistic children/youth, adults and parents) resulted in categories within the domains of body functions (15 categories; referenced 62 times), activities and participation (33 categories; references 160 times) as well as the environment (28 categories; referenced 105 times). The majority of categories were initially linked to third-level categories but re-coded to second-level categories to enable to examine coverage in the autism ICF core sets (see Table 3). A total of 84% of the identified ICF categories were covered by the ICF core set for autism, 19% body functions, 48% activities and participation, and 33% environmental factors. A subset of categories (16%), mostly from the domain environment, was not covered by the ICF core set for autism. See Table 3 for details.

## 5. Thematic Analysis of Focus Group with Swedish Autism Interest Organizations

The thematic analysis yielded 4 themes, encompassing 13 codes. Themes and codes are described below in that order. Themes are denoted in headings. Codes are described in cursive in text and included raw data in quotation marks. See details in Table 4, including frequencies of reporters per code. Quotes are identified by R (respondent) and an identification number. 

### 5.1. Implementation Is Challenging Despite Clear Guidelines 

Participating representatives reported that they were generally *satisfied with information about the guidelines*, one said that “I think that the Public Health Agency’s guidelines that have been available, on their website, they have had one of those checklists you can follow very clearly and great” [R5]. However, they described that while the guidelines were relatively clear overall, members have expressed that *adaptation of guidelines in everyday life* could be challenging. For instance, “There may also be autistic people, older that still lives with their parents and are a bit, have no idea what they should do now. If they should also remain completely indoors” [R1]. Additionally, representatives explained that members have expressed that they have been *worried about how others implement guidelines* as “…our group needs to be separated. We have agreed to follow these rules. But there are others there and then we can’t stay” [R4]. Representatives described that they have been *adapting associations activities for members*, such as “…it’s not like they ask me how to interpret Covid-19 restrictions or the recommendations. If we are having an activity with our members have to try to negotiate. What do you require to dare to attend a member activity? What do you require? Is it ok if we do it like this? Yes, ok. Then we do it that way” [R4].

### 5.2. When Support Structures Diminish, Demands Increase 

Representatives reported a sense from some members of a *sudden loss of everyday support* where one representative stated that “The daily activity centers closed in Stockholm and it was very sudden and without any real preparations” [R2]. Relatedly, several members have reported a *loss of everyday structure*. For instance, “So there are a lot of things that changes and if you are very dependent on your structure in your everyday life everything might easily fall then” [R5] and “It isn’t always easy to figure out what to do instead. In my experience many of our members feel psychologically worse” [R5]. There has also been a sense of increased demands on parents as *parental responsibilities increase* as “Many parents that, parents of children with NDD [neurodevelopmental disabilities] has gotten it really tough. They get no relief now. […] So the parents cannot work at the same time as you have children at home. Even if it is children at high school level for instance” [R5].

### 5.3. Increased Sense of Inclusion in Society

Several reports from members to organization representatives suggested that for some there has been a general sense of further inclusion in society in several respects. For instance, *remote-alternatives increase participation* in schooling and working life, one representative expressed that “Now you can with remote schooling you got approved school-presence in different ways. By for instance login at certain time-points and stuff. After that I know several that have gotten more motivated to complete tasks because then there is still a chance to pass” [R5] and “To transport to work, go by subway, go by train. That might steal the energy of half a working day in itself. So when you don’t have to do that you notice that god no I can, now I can really work” [R5]. Additionally, several representatives expressed a *sense of normalization of lived experiences* where, for instance, “That you, if you work from home and that doesn’t sound weird in any way but it’s, well, it’s because of COVID” [R4] and “That it’s ok to keep your distance and you don’t have to have the forced socialization” [R3]. Indeed, even within the interest organization new forms of communications provided new avenues for member activities following a *digitalization of associations organization*, for instance “I think this will actually permanently change our way of working. It is a pretty significant part and I see that as only beneficial” [R2]. Following development during the pandemic, there is a sense of *optimism for the future*, one representative reported that “Then I’m also thinking about the digital we discussed previously that it has actually entered and developed in society and increase possibilities rather than exclusions” [R3].

### 5.4. Loss of Important Relationships and Prospects for Future

There had been reports from members that there is a *sense of social loss* as “And I think it is pretty significant that some autistic people have been very lonely during all this. It might have been assisted living support, the only person they have seen all week. Got quite a few panic-calls that has been about that” [R1]. Additionally, representatives expressed that there were some that were *worried about the inclusion of the autism community in the future*. For instance, “And I’m afraid that some of what is the welfare and that our members enjoy and create, it helps create their lives, can be threatened” [R2].

## 6. Discussion

The aim of the present study was to inform outcomes in the Swedish autism community in the early phase of the COVID-19 pandemic using a multi-perspective qualitative approach. Interviews with autistic youth and adults were complemented by reports from caregivers as well as representatives of interest organizations across a wide range of life domains. Our study population represents most regions in Sweden although the majority resides in urban areas. Descriptive data show that a variety of co-occurring somatic, psychiatric and neuropsychiatric conditions were present in our sample. Most autistic adults had completed high school or higher education (85%) and the majority of autistic youth attended mainstream schools (74%), albeit with variable supports. Thus, although a breadth of functioning was represented in the study several participants were at the higher end of cognitive and verbal functioning. All participants self-reported practicing social distancing and 23% of autistic adults, 62% of parents, and 71% of youth (16-year-olds or over) were working/studying from home or socially isolated completely. Notably, interviews clarify that even elementary school children (below 16 years old) also socially distanced at home over periods of time as the family self-isolated or due to policies that children should not go to school even with mild symptoms. Participants also clarified experiences that are challenging regardless of pandemic conditions relating to financial concern as well as sleep and food issues. Additionally, as reported elsewhere [14] participants find that accessing satisfactory professional support and care is challenging regardless of pandemic conditions and they appreciate no change in this area. Overall, the formalized pattern of results according to the International Classification of Function, Disability and Health (ICF) [4,38] revealed that close to half of coded outcomes related to activities and participation, about one third to environmental factors, and about one fifth to bodily functions. This overview suggests that interviewed participants experienced that pandemic conditions had pervasive effects on their own health and functioning, on the opportunities for participants to socialize and participate in activities, as well as how the environmental context facilitated or created barriers to functioning. Our multi-perspective qualitative analyses allowed us to synthesize changes during the initial phase of the pandemic that may have contributed to these variable outcomes.

### 6.1. Negative Outcomes

*Mental health*. Participants reported engaging in less health-promoting behaviors, including physical activity, as a part of their typical everyday routines. Additionally, some described disturbed sleep—finding it hard to get to sleep at regular hours, having had issues with getting up in the morning, or loss of sleep hygienic behaviors that promote good-quality sleep. Notably, in accordance with many international studies indicating adverse effects on mental health, the majority of our participants experienced negative mental health effects [16,17,18,19,20,21]. Many reported having low energy and motivation, more worry and low mood as well as difficulties with concentration, and some reported that pre-pandemic mental health and behavioral challenges worsened. Participants mainly related these issues to loneliness, boredom, abrupt and repeated changes, and a general sense of uncertainty. Additionally, the pandemic conditions also induced specific worries about the future such as in terms of job security and fears that policymakers might lose focus on the needs of the autism community when dealing with the crisis. Some also reported increasing worries about transitioning back to pre-pandemic routines and when that might occur. The study findings allowed us to further elucidate factors that may have contributed to the adverse effects of the pandemic conditions on the mental health of our study population.

*COVID-specific concerns*. Findings suggest that issues specifically relating to COVID-19 and associated policies contribute to frustration and worry about general health concerns of infection or affection for someone at risk. While several participants reported that the basic public health guidelines were well understood, adverse emotional reactions relate to implementation in everyday life and the lack of enforcement of them by other people. For children, parents were often able to translate information to everyday rules, thereby reducing concern. However, for some autistic adults, these challenges could be severe, including strict rule-based behavior in the household causing relational distress and significant worries about not adequately following the rules. While some found targeted resources through the Public Health Agency, interest organization, or healthcare providers, helpful this was still a source of distress. Difficulties implementing generalized guidelines into everyday practice is reflected elsewhere as well [27,44] suggesting information dissemination may have not been adequately targeting or been adequately accessible to the population in our study. Several participants reported that the main source of frustration, anger, and worry was the lack of enforcement by other people in the community. These concerns were also relayed to representatives of interest organizations and a factor that impeded structured socializing among the members. Thus, while the pandemic-related guidelines were generally well understood the main sources of concern related to other people’s behavior as well as access to resources to help with the implementation of the guidelines in participants’ everyday life. Additionally, some reported issues pertaining to hygienic precaution, including washing and sanitizing hands, due to sensory challenges of routine changes—findings that align with international studies in all but mask-wearing [44], a policy not enforced in Sweden at this time. 

*Education and work*. Many may have hoped that remote services during the pandemic may reduce specific demands on the autism community, i.e., effortful social interaction, and thus become more tailored for the generalized needs of the population [15]. Our findings corroborate international results in suggesting this hope is only partly fulfilled. Autistic students and employed adults reported that the transition to remote teaching and working could be experienced as sudden, marked by a lack of understanding of specific needs, resulting in loss of planned support for some. Additionally, studying and working from home could be hampered by new distractions, e.g., siblings or children, and several students experienced added communication difficulties including asking for help and comprehending teacher feedback online—in accordance with international studies [44].

A prominent outcome in the present study relates to increasing demands on self-management. Students and employed participants reported having to themselves seek out support, self-initiate work, transition from work to leisure and between activities, sorting through general written information, planning and structuring work, and staying motivated throughout the day to a larger extent during the pandemic. Self-management functions may be particularly challenging for the general autism community [45] and many may need pedagogical support in everyday settings (including home-, school-, and work environments) relating to these challenges [7,8,9]. It is therefore not surprising that our participants found these particular demands caused distress and other negative outcomes. For instance, some students reported not being able to complete the school year as intended. Indeed, teachers report feeling ill-prepared for the rapid change to the new teaching and learning landscape. Although devices and applications may have been available and the pandemic instilled creative and pragmatic solutions from teachers school nation-wide strategies were incoherent, guidelines were not updated according to standards (e.g., GDPR), and experience of use was lacking [36]. Additionally, Huruwitz et al. [26] show that special education teachers in mainstream elementary U.S. schools report innovation and creativity as essential, resulting in changes to intervention content, delivery and format, and worries about the consequences for students when returning to normal teaching formats. Teachers have had difficulties addressing pre-pandemic goals for individual students and have had to rely more on caregivers for intervention implementation while recognizing them not being trained for this activity. Conversely, teachers found that students who may have fared well with classroom teaching were sometimes challenged, and students challenged to participate in classroom teaching sometimes engaged more remotely (Sweden; [36]). Indeed, some participants in our study conversely reported frustration that this intervention became available only when non-autistic students also required it at school. In-person schooling was available to some students, mainly elementary-aged (below 16 years), during this period but young participants and parents reported some similar concerns regarding loss of targeted support. For instance, day-to-day accommodations were abruptly changed or removed, including pick-up and drop-off routines at pre-school, and accommodation to new circumstances could cause significant distress. For employed adults not working from home, workload and accommodations could be set aside as co-workers called in sick or new tasks were introduced pertaining to pandemic-related policies including enhanced hygienic procedures. Due to the timing of the present study, our observations gathered initial crisis responses to the pandemic and associated prevention measures wherefore targeted support may have been re-introduced during later phases of the pandemic.

*Day-to-day life* As the participants in the study were forced to adapt to changes and transitions many reported increased stress relating to self-management of daily routines and structures typically provided by school, work, or leisure activities. While some youth and adults reported making up their own schedules or daily structures and parents coming up with chores and at-home activities for their child many reported that associated with lack of motivation, less energy, low mood, and irritability resulted from this loss of regular structure. Moreover, for some families, safe spaces for outside leisure activities became too crowded with people to visit which further confined families to their home and thus increased this stress. Overall, several participants reported that stressful time spent at home contributed to relationships within the household becoming strained or more severely disrupted. Indeed, international findings report that social service providers have had to adapt the content of their interventions during the pandemic and increasingly target emotional coaching, activation, and parent support [24].

Additionally, while feelings of loneliness can be higher in the autism community in comparison to other groups during pre-pandemic conditions [46,47], participants in our study reported that these feelings increased following reductions in in-person socializing. Notably, however, our findings corroborate international studies to some degree as several participants report loss of specific aspects of socializing including established friendships, family support, and contacts part of established routines at school, work, or leisure activities [27]. These findings suggest a need for strategies to promote social contacts in addition to learning, working, and leisure activities (during isolation) to mitigate adverse outcomes. 

*Care and support*. While some of our participants reported that formal care and support services had been put on hold, including e.g., group interventions and at-home support, and informal support from elderly at-risk relatives, many participants in our study described other challenges to getting support. For instance, added case-loads by case managers at the unemployment office, communication challenges associated with remote support, worries about not being able to socially distance traveling to and meeting in-person, longer waiting periods for at-home food delivery services, or not knowing the contact procedures during the pandemic were among the surrounding circumstances that decreased accessibility to support services. Considering these barriers for support might allow increased access to existing care moving forward. 

*Parental perspectives*. Considering the increased demands on self-management across life areas including everyday routines and studies, changes and barriers to informal and formal support and care services added distress due to COVID-19 itself, and adverse effects on mental health it is unsurprising that parents report added stress and distress in our study. Specifically, parents reported that they struggled to both create and manage their children’s adaptation to new everyday routines. Additionally, many parents struggled to take on the new pedagogical role as an educator for their child. Aligning with findings in international studies [26,31] some parents reported they were not equipped with the pedagogical skills to teach their child at home. Yalmaz et al. [31] found that parents feel helpless with not enough time to adopt new routines following prevention measures and felt lonely and tired in the new role of educator at home. In addition, the support system provided insufficient help coping with the sudden challenges and some parents reported having to quit their jobs to care for their children. Some parents in our study also report that the communication with the school was challenging during the pandemic, including getting adequate information to manage this new responsibility. The added strain on parents is of concern not only in terms of their own well-being but also because of their often essential role in the life of their autistic child [14]. Additionally, studies find that parental health is associated with the emotional and behavioral status of their child [48], which adds to the importance of considering parents moving forward. 

### 6.2. Positive Outcomes

*Social and everyday demands*. Our study supports international findings that pandemic conditions have had some positive outcomes in the autism community by delimiting reductions in specific demands [32,33]. Indeed, several participants reported more energy and less stress. Some reported reductions in social and everyday demands. Specifically, participants describe that pandemic conditions have enforced flexibility in *how* socialization occurs and with *whom* they socialize. For instance, participants could now opt out from face-to-face interactions for school, socialization and leisure activities, work and care/support services, traveling by public transport, engaging in problematic relationships at school, and participating in effortful social small talk. For some, this resulted in more efficient study and work. For some, feelings of inclusion and participation increased as these options became available. Moreover, pandemic conditions *changed* the demands of some typical everyday transitions between activities. For instance, tasks associated with preparations for school and work were reduced while working from home, schedules for work became less cramped, care-related tasks and visits were made available online, and adapting to social and sensory impressions at school was no longer required. As a result, some participants reported more energy and better sleep as well as more time to spend on and develop skills and interests outside of mandatory schooling and working. These findings align with international results highlighting potential benefits of home-schooling, e.g., increased flexibility, reduced distractions, and social demands for some [49]. 

*Normalization of variability*. Pre-pandemic lived experiences of the autistic community may include feelings of stigma, exclusion, and lack of understanding in various areas of life including school [50], workplace [51], professional support and care services [52] and the general community [53]. However, during pandemic conditions, many participants in our study report an overall sense of normalization of their lived experience. Participants found increasingly that not accepting invitations, participating in weekend activities, as well as socially interacting, working, and learning on their own terms was generally better understood. Moreover, several participants reported a hope that this sense of community acceptance is a change that will outlast pandemic conditions and thereby provide an opportunity to participate in society in increasingly variable ways. 

*Access to support*. In addition to a reduction of specific demands, accessible sources of support remained for many or even improved for some. Many participants reported that in-person care and support services remained available—a resource that some report as particularly important as other social connections were lost and new sources of distress developed. For a few employed adults, they were allowed to do more work-related activities that suited their proclivities during pandemic conditions. For students going to school or pre-school, changes to in-person routines were actually specifically targeting their needs, such as smaller-sized peer groups and spending more time outside during breaks at school. More time spent at home also allowed some household relationships to strengthen, becoming both a way to cope with change and unpredictability and also a strengthening source of support. Some families reported they had more flexible time together, more energy, and less stress. 

*Coping with changes*. Several aspects of this study also highlighted other ways in which participants were able to cope and adapt during pandemic conditions. Participants reported that while remote socialization and participation in leisure activities may not be the preferred formats, it was nevertheless a way to maintain connection and rewarding activities. Moreover, some students transitioning to remote teaching similarly conceded that the new format may not be optimal and associated with several challenges. However, a sense of trust by one or several school staff along with practical support that targets individual needs, including regular check-ins regarding both learning and well-being, and the experience that teachers made conscious efforts to provide information and allow predictability as far as possible was associated with fewer feelings of distress. Additionally, several participants also reported a sense of optimism looking to the future, in terms of how the pandemic conditions may create opportunities to learn and society to improve, acceptance, in terms of the transient conditions’ status during the pandemic, as well as functional strategies made available by past experiences relating to similar demands. These strategies included ways to create daily structures, therapeutic strategies to promote mental health, and finding new ways to perform health beneficial behaviors including staying physically active by taking regular walks. Such coping mechanisms closely relate to research on generalized factors promoting resilience and may become fruitful avenues for interventions to mitigate negative long-term outcomes in the autism community following COVID-19.

Thus, overall, participants specify adverse outcomes to multiple areas of life. Negative outcomes include pervasive changes unpredictability relating to daily routines and activities, new challenges associated with new formats of education, work, care, and socializing, loss of opportunities to socialize, increasing demands on self-management, barriers to care and support, new sources of distress relating to the implementation of pandemic-related prevention measures, and depletion of parental resources. In turn, participants report notable detriments to the overall mental well-being, including increased anxiety, low mood and motivation, as well as household relationships. On the other hand, participants also reported some positive outcomes including specific reductions in pre-pandemic demands in everyday demands, including social demands and everyday transitions, as well as new opportunities to socialize and participate in activities and support services, received targeted support, and strengthened familial relationships. More generally, the pandemic conditions enforced flexibility, options, and an increased sense of acceptance and understanding to the benefit of participants in our study. In turn, participants reported more energy and less stress. Following these findings, policymakers may consider adopting strategies to promote optimal outcomes including further developing common resources for targeted pedagogical and support during remote schooling and working, provision of changing formal care and support needs and potential barriers to accessing these services, increasing targeted parental support, and opportunities for the autistic community to participate in safe, structured activities. Policymakers may also consider mitigating potential long-term outcomes by maintaining viable options that have promoted flexibility and targeted support during the pandemic as well as promoting delineated coping mechanisms outlined here. 

The interpretation of the study results should be made with some limitations in mind. Firstly, the present study was circumscribed to the initial stages of the pandemic and potential generalization to later stages—marked by changes in restrictions and COVID-19 knowledge—is limited. A longer-term follow-up might contrast the findings based on the earlier experiences of the pandemic. Secondly, the study aimed to increase understanding of the lived experience of the autism community specifically and cannot argue potential differences of findings to other comparison groups. Quantitative studies may elucidate comparative differences such as the degree of impact in various life areas. Thirdly, albeit having separated out parent experiences, the scope of the present article does not include age-group comparison. This decision was based on the observation that overarching themes could often defy an age range. Nevertheless, focusing on higher-level thematic patterns of the autism community as a whole might miss specific outcomes pertaining to the diversity of sub-groups represented within the community. Future analyses may specifically analyze areas of life or age-specific experience in further detail or discrepancies between, e.g., child and parent dyads, which may further nuance the understanding of lived experiences in various settings and sub-groups. 

## 7. Conclusions

While the COVID-19 pandemic and related national prevention measures have had catastrophic effects on both individual and societal levels, these extraordinary circumstances may allow opportunities to enhance our understanding of the lived experiences of the autism community at this time and beyond. The present study sought to inventory the lived experiences of the Swedish autism community through multi-perspective qualitative analysis during the initial phase of the COVID-19 pandemic in order to guide intervention planning and preparedness for future pandemic or epidemic outbreaks. Formal inventory of patterns of results using the ICF revealed (i) pervasive effects of factors relating to the function and well-being of the individual but notably changes to participation in activities as well as delineating environmental facilitators and barrier, and (ii) that the ICF core sets for autism were generally able to cover the lived experience of our study population. Specifically, results revealed declining mental health, challenges to implementing governmental guidelines, barriers to access support and adequately participate in education, work, socialization, and leisure activities as well as depletion of parental resources at this time. Conversely, finding elucidate coping mechanisms and better-than-expected outcomes in terms of specific demand reduction and normalization of everyday lived experience. While only reflecting the initial phase of the pandemic in Sweden, adverse outcomes likely have lasting effects that require consideration and beneficial effects of pandemic conditions may be opportunities to improve the lived experience through and beyond the pandemic. Future studies may enhance understanding by group analyses, long-term follow-up, as well as cross-country comparative studies which may reveal discrepancies in outcomes for the autism community following variability in national prevention measures to increase our understanding of how comparative challenges can be met in the future.

*Authors note*: The study uses identity-first language as studies indicate this may be the preferred language by members of the autism community [54].

## Figures and Tables

**Table 1 ijerph-19-01268-t001:** Characteristics of participants (excluding representatives of autism interest groups).

	Autistic Adults (n = 13)	Parents of Autistic Children (n = 13)		Young People ^2^ (n = 7)
		Autistic Parents ^1^ (n = 2)	Non-Autistic Parents (n = 11)	
**Age in years**	M = 33.38 (SD = 8.00) Range: 26–55	M = 38.5 (SD = 0.71) Range: 38–39	M = 45.64 (SD = 4.99) Range: 39–53	M = 16 (SD = 1.41) Range: 14–17
**Gender**	Woman: n = 10 Man: n = 2 Other: n = 1	Woman: n = 2 Man: n = 0 Non-binary: n = 0	Woman: n = 8 Man: n = 2 Non-binary: n = 1	Girl: n = 3 Boy: n = 2 Non-binary: n = 1 Other: n = 1
**Place ofresidence**	Stockholm: n = 7 Skåne: n = 1 Västernorrland: n = 1 Blekinge: n = 1 Örebro län: n = 1 Östergötland: n = 1 Prefer not to say: n = 1	Stockholm: n = 2	Stockholm: n = 7 Västra Götaland: n = 2 Skåne: n = 2	Stockholm: n = 4 Västra Götaland: n = 1 Örebro län: n = 1 Skåne: n = 1
**Number of people in household, including self**	1: n = 7 2: n = 3 3: n = 3	1: n = 0 2: n = 0 3: n = 1 4: n = 1	1: n = 0 2: n = 2 3: n = 4 4: n = 3 5: n = 2	1: n = 0 2: n = 1 3: n = 3 4: n = 2 5: n = 1
**Predominant ethnic background**	European: n = 13	Latin American: n = 1 European: n = 1	Latin American: n = 1 European: n = 9 Asian: n = 1	European: n = 7
**Highest qualification/current school placement**	University degree: n = 2 Vocational educational training: n = 2 Military college: n = 1 Community college: n = 1 Completed high school: n = 5 Completed primary school: n = 1 Started primary school: n = 1	University degree: n = 1 Vocational educational training: n = 1	University degree: n = 10 Vocational educational training: n = 1	Mainstream with no extra support: n = 1 (from transcript) Mainstream with extra support: n = 3 (one form trancript) Special school: n = 3 (2 from transcript)
**Current** **occupational status**	Part-time employment: n = 1 Full-time employment: n = 3 (Parental leave: n = 1) Studying: n = 5 (of which Part-time: n = 2) Unable to work due to disability: n = 4	Part-time employment: n = 1 Self-employed: n = 1	Part-time employment: n = 3 Full-time employment: n = 7 Studying: n = 1 (of which Part-time: n = 1) Unable to work due to disability: n = 1	
**Participant co-occurring developmental, psychiatric, and medical conditions** (Past and current)	ADHD: n = 7 Anxiety disorder: n = 1 Depression: n = 3 Tourette syndrome: n = 1 PTSD: n = 2 Allergy: n = 1 Asthma: n = 1 OCD: n = 2 Migraine: n = 1 Personality disorder: n = 1 Psoriasis: n = 1	Anxiety disorder: n = 1 Asthma: n = 1	ADHD: n = 1 Chronic fatigue syndrome: n = 1 Eye melanoma: n = 1 Frozen shoulder: n = 1 Migraine: n = 1 Parkinson’s disease: n = 1 Thyroidectomy: n = 1	ADHD: n = 2 Anxiety disorder: n = 1 Cerebral palsy: n = 1 Depression: n = 1 Ehlers-Danlos syndromes: n = 1 Language disorder: n = 1 POTS: n = 1
**Confirmed or suspected COVID-19 infection [self (in household)]**	3 (1)	0 (2)	0 (2)	1 (0)
**Degree of social distancing/self-isolation**	Social distancing: n = 10 Social distancing and working from home: n = 1 Self-isolation: n = 2	Social distancing and working from home: n = 2	Social distancing: n = 5 Social distancing and working from home: n = 4 Self-isolation: n = 2	Social distancing: n = 2 Social distancing and studying from home: n = 3 Self-isolation: n = 2
**Number of autistic children**		n = 2	n = 14	
**Children’s age**		M = 7 (SD = 1) Range = 6–8	M = 14.2 (SD = 2.8) Range = 8–17	
**Children’s gender**		Boy: n = 2	Girl: n = 1 Boy: n = 5 Non-binary: n = 3 Prefer not to say: n = 1	
**Children’s residence**		With two parents: 2	With two parents: 7 With one parent and one step-parent: 1 Shared parenting: 1 With one parent: 2	
**Children’s autism diagnoses**		Autism spectrum disorder: n = 1 Atypical autism: n = 1	Autism spectrum disorder: n = 4 Asperger’s Syndrome: n = 4 Pervasive developmental disorder: n = 1 Atypical autism: n = 1	
**Children’s co-occurring diagnoses** (Past and current)		Hearing loss: n = 1	None: n = 1 ADHD/ADD: n = 6 Anxiety disorder: n = 2 Cerebral palsy: n = 3 Depression: n = 2 Dyspraxia: n = 1 Hearing loss: n = 1 OCD: n = 1 Language disorder: n = 2	
**Children’s current school setting**		Mainstream with no extra support: n = 2	Mainstream with extra support: n = 6 Autism-specific class within a mainstream school: n = 2 Special school: n = 2	

^1^ Two parents of autistic children had autism themselves. Their information is listed in a separate column, in order to make it possible to read in conjunction with the autistic adults’ column. ^2^ Two young participants’ parents also participated. Some information for these two participants is therefore listed both in the young participants’ column and in the parent’s columns.

**Table 2 ijerph-19-01268-t002:** Characteristics of representatives of autism interest groups.

	Representatives of Autism Interest Groups
**Age in years**	M = 47.40 (SD = 4.89) Range: 30–72
**Gender**	Woman: n = 2 Man: n = 3
**Place of residence**	Stockholm: n = 3 Västernorrland: n = 1 Örebro län: n = 1

**Table 3 ijerph-19-01268-t003:** Overview qualitative analyses and ICF linking of interviews with autistic participants and caregivers.

Theme	Subtheme	Codes	Example Units	ICF Links
**HEALTH DURING COVID**	DETRIMENTS TO OVERALL HEALTH	**Understimulation cause distress** **(N = 7)**	“at the same time I’ve also had a hard time finding motivation for school and sort in general. And also kind of, like, I don’t know, a sense of meaninglessness, a bit understimulated in general” [YP-78] “What he needed then was rest. But now he is well rested. And the you notice that he is a bit too bored again” [PP-67]	* b130 Energy and drive functions * b152 Emotional functions *** e555. Associations in organisational services, systems and policies
		**Restrictions take a toll on mental health** **(N = 22)**	“You now, the mental health has been affected. I’s kind of lowered. Cause it’s like, I don’t like change and the loneliness and the uncertainty” [YP-73] “I usually have a lot of worries and anxiety but I think it has become worse” [AP-01] “I easily get, I get low to put it simply. It get’s harder to do things..” [AP-12] “I have always had sleeping problems that have now become even worse” [AP-14] “He has slit opened his wrists at one time during the spring. It was a situation where he was also feeling more anxious” [PP-13]	* b152 Emotional functions * b130 Energy and drive functions * b160 Thought functions * b134 Sleep functions * e570 Social security services, systems, and policies * b765 Involuntary movement functions * d920 Recreation and leisure * d570 Looking after one’s health * d760 Family relationships * d770 Intimate relationships * d240 Handling stress and other psychological demands * d230 Carrying out daily routine
		**Continuous health concerns** **(N = 18)**	“Because it’s also that if I would get COVID, the fever from COVID could make me have another attack” [YP-73] “And he was worried, will you die mummy?” [PP-70]	b152 Emotional functions * b160 Thought functions d240 Handling stress and other psychological demands * d570 Looking after one’s health e310 Immediate family
		**Continuous financial concerns** **(N = 6)**	“But I definitely have financial worries all the time. Because of COVID-19, amongst other things” [AP-06]	* b152 Emotional functions * d870 Economic self-sufficiency e570 Social security services, systems, and policies e580 Health services, systems, and policies
		**Less daily activity** **(N = 14)**	“But I have been outside a dozen times at most since March. That might not be super healthy” [AP-06] “But I notice that I’ve gained some weight. And I think that might be because I haven’t gotten that natural everyday exercise since COVID-19 come along” [AP-32]	* b130 Energy and drive functions * b152 Emotional functions * d570 Looking after one’s health * d920 Recreation and leisure ** b530. Weight maintenance functions
	SUPPORTED WELL-BEING	**Thriving during the pandemic** **(N = 16)**	“I have a lot more energy left to do things I like during the week. And that’s because I kind of don’t need to be in the classroom where there can be a lot of noise that can disturb me” [YP-08] “you know, I’ve been very well actually, surprisingly well actually, it probably also has something to do with not stressing myself to be active all the time” [YP-69] “then I don’t leave the house because I find it hard to deal with large crowds. But now I know it’s often kind of empty. So then it’s great like, I get out more” [AP-33] “Now she has actually started doing it. Now during the summer vacation she’s been away and they have met several times. The previous summer she didn’t meet anyone for the entire summer” [PP-07] “He is not scared of the disease. He has no interest in the news” [PP-67]	* b152 Emotional functions * b126 Temperament and personality functions * e250 Sound * b130 Energy and drive functions
		**More time for interests/talents/passions** **(N = 14)**	“I draw as I usually do. But a bit more now that I am home” [YP-11] “But that you realize, damn I can do more than I thought. That you have more time to actually immerse yourself in something and get better at it” [AP-06]	* b130 Energy and drive functions * b160 Thought functions d650 Caring for household objects * d920 Recreation and leisure
		**Found other ways to stay active** **(N = 9)**	” Well, the physical is probably about the same. Potentially my conditioning is slightly better because I walked outside a bit more than I usually do” [YP-78] “But then I felt that—no this is going to last for a while so then I will leave. Because I can live more actively here, and it is easier for me to follow restrictions and stuff like that” [AP-32] ”Since you haven’t been as much with friends I think my judgement is that we might have been even more out in the woods” [PP-75]	* d570 Looking after one’s health * d920 Recreation and leisure ** e298 Natural environment and human-made changes to environment, other specified
		**My pet takes me outside** **(N = 5)**	“She [cat] keeps me alert and activated. We play a lot indoors and I usually walk with here outside in a leash in the woods area behind the house where I live” [AP-14]	** e350 Domesticated animals ** e298 Natural environment and human-made changes to environment, other specified
**ACCESS TO REGULAR SUPPORT**	RESTRICTED ACCESS TO SUPPORT	**Regular support services have become unavailable for us** **(N = 20)**	“I have also been a bit unsure of how things work with the habilitation services and that kind of stuff, if you can go there or if it is over the phone and that kind of stuff. So I haven’t been there all spring either” [AP-05] “I never got a first meeting. And already then the unemployment agency was under a lot of pressure anyway cause they receive less money, so that probably also has an effect. But then COVID came along and there are more unemployed and the unemployment agency still have little money and I heard that my administrator had 160 unemployed that only she was in charge of (laughter)” [AP-31] “travelling by public transport or sitting at a healthcare service space with other people [and talking online while] partner was working from home” [AP-12] “So you haven’t been able to, I haven’t been able to meet my parents either. So there hasn’t been any respite so” [AP-65] “And ordered home delivery so you get it to the door. It has worked great. But from March, April and the May it was three and a half week, the delivery times where prolonged. And that didn’t work for me. Because I don’t have the ability to look that far ahead” [AP-35]	d730 Relating with strangers * d360 Using communication devices and techniques e580 Health services, systems, and policies e315 Extended family e355 Health professionals e340 Personal care providers and personal assistants * d620 Acquisition of goods and services * d230 Carrying out daily routine d330 Speaking
		**Remote support services offer added communicative challenges** **(N = 5)**	“Yes it’s different because you cannot sense the mood as well over the telephone. It can also be difficult for me to know when the other person has stopped talking, or if they have just taken a break when I talk on the phone. That I might interrupt even though that’s not my intention. And I’ve noticed that she is a bit more disorganized like that” [AP-32] “Just thought it was hard and weird [video-call with psychiatrist]. When it’s about that type of stuff it gets strange” [PP-67]	* d360 Using communication devices and techniques d330 Speaking * d315 Communicating with—receiving—nonverbal messages e310 Immediate family
		**Finding adequate support is always difficult for us** **(N = 14)**	“No. I hold it together yeah…It is not sustainable. I know it’s not sustainable. But it is what I got” [AP-12] “I don’t really think I got that support before COVID either (laughter). I have mostly been shuffled around different people or been told ‘no’ cause I have autism and they don’t deal with that, or they don’t know anything about that” [AP-31]	* d760 Family relationships b122 Global psychosocial functions * b125 Dispositions and intra-personal functions * b130 Energy and drive functions d210 Undertaking a single task d220 Undertaking multiple tasks * d230 Carrying out daily routine e310 Immediate family e330 People in positions of authority e340 Personal care providers and personal assistants e355 Health professionals e360 Other professionals e580 Health services, systems, and policies e585 Education and training services, systems, and policies
	ACCESS TO REGULAR SUPPORT	**Remote support allows flexibility** **(N = 14)**	“We have talked over Teams Speak […] Ok. Easier to talk […] That it’s faster […] That we don’t have to go there and sit in like a waiting room and go back” [YP-76] “For me, this works much better, mostly cause then you don’t have to like, like she comes quite early in the morning, and then if it’s over the phone I can sleep a bit longer and stuff. So that feels much better” [AP-33] “Then after, a few weeks, maybe one month passed, they could offer the course online. That was wow. Why haven’t you done this earlier. It was very positive” [PP-70]	* e125 Products and technology for communication * d360 Using communication devices and techniques d730 Relating with strangers e355 Health professionals ** e398 Support and relationships, other specified
		**In-person support has been available as usual** **(N = 16)**	“Yes, that I can call when I feel bad. I can meet in person more often if I want to. It’s kind after what I need. I appreciate that enormously” [AP-01] “Talk a lot with my housing support person about my anxieties. They usually help me to go to the grocery store and sometimes go for walks together […] My housing support has been especially important now” [AP-14] “Yeah, we’ve continued as usual [with support family] even with cold symptoms and stuff like that” [PP-02]	e340 Personal care providers and personal assistants e355 Health professionals
**NEED FOR SOCIALIZING**	SENSE OF LOSS OF SOCIAL CONNECTION	**Less in-person socializing** **(N = 15)**	“But yes, I have kept in contact with a few a little. Not as much as I usually do. And even though I have more energy now it is still nice to spend that energy on being with people. Because I notice I need that” [YP-78] “I guess it’s one or two that live close by where I live. I’ve been able to see them. I’ve biked to one of them. […]. And so it’s been outside then” [AP-01] “Some of his friends that lives in other regions or they, they live in [the city], we have, we simply don’t play with them. So that’s kind of sad” [PP-74] “thinks it’s sad that he cannot meet the friends that aren’t in the same pre-school. In some way we have drawn a line there as parents” [PP-70]	* d750 Informal social relationships *** e545 Civil protection services, systems and policies
		**Missed in-person connection with family and relatives** **(N = 12)**	“I have reacted quite negatively to that I think, because my grand-father dies so I would have wanted to meet him before and kind of comfort my grand-mother” [YP-08] “He misses his grand-father very much. And when you talk, it is not possible now because it’s corona-times. So it’s mostly like ‘I hate corona, I hate corona’” [PP-13]	* d760 Family relationships e310 Immediate family e315 Extended family
		**Social recreational activities have stopped** **(N = 15)**	“I had, I really don’t have any close friend so but I have many acquaintances in Pokémon-Go that I meet a lot when I’m out playing, and I have barely seen them at all now” [AP-31] “Previously I was active in [interest organisation] but now I’m not active in anything. I don’t feel a part of the community at all” [AP-14] “Yes, especially swim practice, that hit him hard. Because he loves to swim” [PP-13] “But now we’ve ordered, all of his clothes online. We’ve been out shopping for shoes one single time. […] That’s the only time he has been out for this entire period of time” [PP-75]	* d570 Looking after one’s health * d920 Recreation and leisure * d910 Community life e320 Friends e325 Acquaintances, peers, colleagues, neighbors, and community members
		**Feeling lonely** **(N = 10)**	“But then it’s more about the loneliness. Because I don’t have many good friends either. That I have contact with. So it’s just been like I’m a bit lonely here. I have my parents but they aren’t a substitute for friendships” [YP-73] “been going pretty hard core self-isolation, so kind of like on a USA-level. We’ve hardly gone outside. So it’s been very lonely” [YP-69] “It has affected me in the sense that I might feel a bit more lonely than I would have otherwise, cause it’s, social activities have disappeared” [AP-33]	b122 Global psychosocial functions * d760 Family relationships * d920 Recreation and leisure e310 Immediate family e315 Extended family e320 Friends
		**Remote connection fall short** **(N = 17)**	“I’ve had piano-lessons. Which has been over FaceTime instead […] At the same time it’s a bit harder to get feedback in different ways because it is something you do physically with your hands. I don’t know. Bu yeah, kind of neutral” [YP-78] “It is like drinking Cola Zero instead of regular Coke when you are a sugar addict. You kind of get the right feeling but you know it’s still not quite right” [AP-65] “I don’t know, I don’t want to call too much because then she [grand-mother] thinks we are in [her country]. And then she goes looking for us upstairs” [YP-08] “But I don’t like it at all. I think it disturbs my brain very much” [AP-05]	d720 Complex interpersonal interactions * d750 Informal social relationships d760 Family relationships * d360 Using communication devices and techniques * d920 Recreation and leisure e360 Other professionals * e125 Products and technology for communication
		**Misses incidental contacts in regular structured environments** **(N = 8)**	“Yes, I have missed that actually, we usually sit at the same table every lunch” [YP-11] “He has missed the social in school too. He has. He has said that. And it is noticeable too” [PP-77] “There are very clear rules there and everybody talked to each other. I don’t think there was anyone who did not talk to someone else. So yeah, it was a bit of both. Miss the colleagues” [AP-65]	* d720 Complex interpersonal interactions * d750 Informal social relationships
	SOCIALIZING ON MY OWN TERMS	**Normalization of social distancing** **(N = 6)**	“Maybe not tomorrow or the day after tomorrow, change plans and stuff like that cause I don’t have the energy. But now I haven’t had to explain myself all the time. Instead no one has asked and I haven’t explained” [AP-12] “I have tried to use the COVID-epidemic to explain to others what my everyday life usually looks like” [AP-35] “We usually make jokes about it and say that it will be this family that thinks that this has been good. We are already experts on this social distancing. So we haven’t had any problems with that part” [PP-07]	e420 Individual attitudes of friends e460 Societal attitudes ** e425 Individual attitudes of acquaintances, peers, colleagues, neighbors and community members
		**Fewer demanding social contexts** **(N = 16)**	“Now that I don’t, now when I haven’t been in school I haven’t for instance travelled by public transport, I haven’t spent time with people. And then I’ve gotten a bit more energy” [YP-78] “I probably had some difficult relationships in school before all this so this was a good opportunity to kind of a bit naturally drift away from them, so that was actually kind of nice” [YP-69] “Because the positive is that you don’t have to travel and stuff like that. That you have been able to do a lot digitally instead, for example work and stuff like that” [AP-06] “I think it’s been pretty nice. I don’t want any close contact with people. I don’t like talking to other people unless I have a reason to” [AP-35] “That you don’t have to be at work and spend a lot of ineffective hours at work to have breaks with others and you’re supposed to socialize and talk, which I don’t like. But you have to if you want to keep your job” [PP-70]	b122 Global psychosocial functions * b130 Energy and drive functions b265 Touch function d240 Handling stress and other psychological demands * d720 Complex interpersonal interactions d730 Relating with strangers d740 Formal relationships * d750 Informal social relationships * d770 Intimate relationships d820 School education d850 Remunerative employment * d920 Recreation and leisure
		**Online socializing can fill my social needs** **(N = 16)**	“Before I could meet them physically, which I haven’t done as much now. I have been able to video-talk and stuff with them and stuff, still have conversations. And that allows me to have much more energy to actually talk to them” [YP-08] “It’s been pretty nice that you can finish when you get tired. You don’t have to wait until the other person goes home or anything like that” [YP-78]. “I kind of think it doesn’t matter. As long as it’s fun” [YP-76] “before we saw each other, or we have not known each other that long but now it’s only telephone calls. But that feels good. Because it’s important to me as it were” [AP-01]	* d360 Using communication devices and techniques * b122 Global psychosocial functions
		**In-person socializing with family and relatives** **(N = 10)**	“like, it has worked. But then before the Corona pandemic started I was on sick leave due to depression. And that was very difficult. But then we did, but she [mother] came and saw me anyway because it kind of was needed because sometimes I feel so bad I can’t be on my own” [AP-33] “But she herself said to me, wow wow I forgot how to small-talk. That’s great I said, then you can practise a bit on my mother then, that came” [PP-09-10] “Grand-father used to come to our home here outside and into the garden and say hi through the window. So we started a little game, garden cafe. So they got to come by and got a coffee through the window and that kind of stuff” [PP-70]	* d920 Recreation and leisure e310 Immediate family e315 Extended family
**PARTICIPATION IN SOCIETY**	RESTRICTED PARTICIPATION	**When others implement restrictions my participation becomes limited** **(N = 11)**	“he absolutely cannot be outside if it’s windy, and the sun is too hot and stuff like that, so then the weather has to be pretty perfect if it feels ok enough for him to go outside you know” [PP-02] “So that sometimes it was Wednesday, then it was on a Tuesday, then there was another practice and then they were practicing together with another group and then they were going to be outside, then inside, then it was going to be on a Wednesday, yeah” [PP-07] “He and I have been bathing quite a bit from the pier. But then it got crazy hot, then this carer got here and she has done that with him. Then it got crazy hot and it got really crowded. So he couldn’t go bathing with him anymore” [PP-13] “if you maybe usually go to a forest every week then sometimes there have been so many people so you have had to skip it, and maybe go to places further away or adapt in some other way” [AP-05] “Because it is embarrassing. It feels so associated with so much guilt in some way and as soon as you cough, someone is looking at you. I can’t really deal with that” [PP-70]	* b270 Sensory functions related to temperature and other stimuli * d750 Informal social relationships e320 Friends * b125 Dispositions and intra-personal functions * d220 Undertaking multiple tasks * d230 Carrying out daily routine * d920 Recreation and leisure d730 Relating with strangers b160 Thought functions * d570 Looking after one’s health *** b210 Seeing functions
	PARTICIPATION WAS MADE POSSIBLE	**In-person continued participation in recreational activities** **(N = 3)**	“I feel like it’s necessary for my health in some way. Cause I think I would almost go crazy if I couldn’t” [AP-01] “he was signed up for a climbing course. And they hadn’t cancelled that and we actually went to it. Cause when I spoke to the trainer everyone had cancelled their participation except for our son” [PP-13]	* d570 Looking after one’s health * d920 Recreation and leisure
		**Online alternatives can allow participation in recreational activities** **(N = 6)**	“Yeah, we had some online-practices that were easiest to attend; it wasn’t like you had to get ready for like an hour beforehand” [YP-08] “when they had online-practices she was able to attend anyway. I thought that worked relatively well. And it was a bit exciting because her trainer was quarantining in [another country]” [PP-07]	* d920 Recreation and leisure * e125 Products and technology for communication * d910 Community life
**ESTABLISHING CIRCUMSTANCES WHEN MOVING ONLINE**	NEGATIVE EXPERIENCES	**Lack of understanding and adequate support** **(N = 9)**	“I think we got the e-mail on the evening of the Wednesday, that it [assistance participating in daily activities] would be closing down from Monday…since it was remote-learning I had been studying at the [assistance participating in daily activities] ” [AP-04] “I remembered, there was another small change, or a pretty major change in all this, [child’s name]’s favourite teacher suddenly went on sick leave from school” [PP-77] “The whole thing in this situation got, it was brought to its head now that COVID came along. Because suddenly when all the typical kids or whatever you say, that have no diagnoses, when they suddenly were at home, support was somehow arranged for how you can handle it when kids aren’t really coming to school. For [child’s name] couldn’t come to school alone before COVID because he was too afraid and refused to go if none of us [parents] came along” [PP-67]	e130 Products and technology for education e330 People in positions of authority e525 Housing services, systems, and policies e590 Labor and employment services, systems, and policies * d820 School education e360 Other professionals e430 Individual attitudes of people in positions of authority
	POSITIVE RELATIONSHIPS AND COMMUNICATION	**Trust in staff** **(N = 7)**	“So I’ve had good teachers and stuff. But I’ve still had a hard time. And they have understood that. But I still want to celebrate my teachers because they have been awesome at this” [YP-73]	* d820 School education d850 Remunerative employment e330 People in positions of authority e360 Other professionals
		**I felt prepared as we transitioned** **(N = 10)**	“There were clear routines for how to do it too pretty early. There were weekly updates from the company board regarding how we should try to relate to things” [YP-73] “since like all at that school has autism the teachers get a lot. They have a lot of experience and understand how we work. So it has been easier to adjust from that and, yeah, general understanding… They give us information in time, they, I know they think a lot about that” [YP-78] “So, well the usual then, since it was remote-learning from the beginning it, well, continued to be remote in all sort of courses and high school and then stuff started to do remote learning. So, there wasn’t any changes and stuff” [AP-04]	d850 Remunerative employment e330 People in positions of authority b152 Emotional functions * d230 Carrying out daily routine * d820 School education * d830 Higher education * e125 Products and technology for communication e360 Other professionals
**LEARNING AND WORKING ON SITE**	LEARNING AND WORKING BECAME HARDER	**In-person routine changes fail to attend to individual needs** **(N = 11)**	“And he has thought that it’s been really hard that, for instance, we have to leave and pick him up outside of the pre-school all of a sudden. That has been a really hard change. And it’s a lot of those kind of things I primarily haven’t been able to kind of control” [PP-02] “It might have been a bit inflexible there, initially” [PP-74] “Those that have come back, including me, had to work double after. So no, I can’t say that. Nothing positive, the opposite. More stress I would say” [AP-12]	* d850 Remunerative employment e325 Acquaintances, peers, colleagues, neighbors, and community members b152 Emotional functions * d230 Carrying out daily routine d240 Handling stress and other psychological demands * d820 School education e325 Acquaintances, peers, colleagues, neighbors, and community members e360 Other professionals
	LEARNING AND WORKING WAS FACILITATED	**Did more of what I’m good at** **(N = 3)**	“Because of that they know that I am, it is part of my autism-superpower kind of that I am skilled at administrative work. So, I’ve gotten more hours of that. Which has been very nice anyway. Because then my income isn’t zero” [AP-06]	d850 Remunerative employment e325 Acquaintances, peers, colleagues, neighbors, and community members
		**In-person routine changes target individual needs** **(N = 4)**	“No but we were sitting at the same seats as… But those that were kind of sick got to sit and like wait on Google Meet. And there’s a teacher that show what they were supposed to do. In school” [YP-76] “And I think that, that the group of children has gotten smaller [due to illness at kindergarten] has made things easier for him socially and not made him so tired and exhausted” [PP-02] “before they only went outdoors during the lunch break that is a bit longer, now they go outdoors during all of the breaks. But he likes to go outside so I think he thinks it’s nice. And outside the volume is a bit lower and he can keep to himself if he wants to” [PP-66]	* d820 School education e360 Other professionals
**MEETING THE CHALLENGE OF REMOTE LEARNING AND WORKING**	SPECIFIC BARRIERS OF REMOTE LEARNING	**Remote learning prevented targeted support** **(N = 6)**	“You could say that is one negative thing about school since the lessons often goes a bit long and the next lesson starts almost a few minutes early. So there is no break at all between” [PP-07] “They haven’t been able to give, provide adequate support either to be able to structure the school day for [child’s name]. And been able to have the kind of follow-up that might have been needed to push in the right direction on the right stuff” [PP-09-10]	* d220 Undertaking multiple tasks * d230 Carrying out daily routine * d360 Using communication devices and techniques * d820 School education e325 Acquaintances, peers, colleagues, neighbors, and community members e360 Other professionals
		**Remote learning and working challenge communication** **(N = 9)**	“Sometimes their microphones isn’t working or it’s lagging, it’s hard to kind of have the time to do everything or say what to do when the microphone is off” [YP-11] “it’s a bit harder to get feedback I guess. The teacher can’t see your face so it gets harder for them to interpret what you want or feel and things like that. And many, especially at my school, and me, have some difficulties asking for help and formulating what you need help with” [YP-78]	b164 Higher level cognitive functions * d210 Undertaking a single task * d310 Communicating with—receiving—spoken messages * d335 Producing nonverbal messages * d360 Using communication devices and techniques * d820 School education d850 Remunerative employment * e125 Products and technology for communication e360 Other professionals ** d345 Writing message
		**Remote learning increase demands in self-management** **(N = 17)**	“It’s been harder to motivate myself for school because the boundary between spare time and school has been a bit erased” [YP-08] “When COVID-19 hit everything was upended. All physical meetings were cancelled. Now everything had to be managed online and I was overwhelmed with course material/online-seminars and similar” [AP-14] “That is one of the largest difficulties for her. To manage things on her own. Planning and structuring her own work and finish” [PP-09-10] “I feel insufficient when I see that there is still this much to finish in my e-mail inbox and I have only this many hours to work. I have to finish now. It is really hard for me to just wrap up and leave unfinish” [PP-70]	* b164 Higher level cognitive functions * b140 Attention functions * d160 Focusing attention e330 People in positions of authority d330 Speaking
		**Distractions while working and studying from home** **(N = 5)**	“Hard, because you can’t sit where you usually sit for instance, I usually sit in the kitchen, then my siblings are running around and disturb me. They run around and shout and play” [YP-11]	* b140 Attention functions d160 Focusing attention d161 Directing attention * d720 Complex interpersonal interactions * d820 School education d850 Remunerative employment *** b230 Hearing functions
		**Can’t follow my study plan** **(N = 5)**	“It hasn’t gone so great, I’ve had to remove several subjects but now I will re-do first year at high school because I wasn’t able to finish what I had to” [YP-11]	d820 School education * d360 Using communication devices and techniques * d315 Communicating with—receiving—nonverbal messages * b130 Energy and drive functions * b140 Attention functions * d160 Focusing attention e330 People in positions of authority
	SPECIFIC FACILITATORS OF REMOTE LEARNING AND WORKING	**Regular check-ins** **(N = 9)**	“And the teachers have been, but always, very often surveys to see how we’re doing. And then said that if you have any questions or anything to just keep in touch” [YP-73] “It is the first time that ever that I have seen a detailed plan saying that this is what you’re supposed to do” [PP-67] “Information-flows, clear and clear structure. Generally, they have handled it very well from school” [PP-77]	d240 Handling stress and other psychological demands * d360 Using communication devices and techniques d740 Formal relationships * d820 School education e360 Other professionals
		**Studying is more efficient from home** **(N = 7)**	“It’s not as effortful, so it’s easier to ask the teachers questions” [YP-08] “Yes, he has been able to add a subject and remove another this year” [PP-77]	* b130 Energy and drive functions * b152 Emotional functions * d360 Using communication devices and techniques d330 Speaking d132 Acquiring information * e250 Sound e310 Immediate family
**PARENTAL RESOURCES DURING COVID-19**	ADDED STRESSORS	**Communicative challenges between school and parents** **(N = 4)**	“What we’ve had to do is to is to try to have communication with the pedagogues at, the special education pedagogues, teachers and principals and stuff at school to an extent. But they have had very little capacity too” [PP-09-10] “And gotten some confirmation if everyone in the team that helps with his needs but may not be his ordinary teachers, are aware of the extra support he gets and stuff ” [PP-66]	* d820 School education e330 People in positions of authority e360 Other professionals
		**Added pedagogical responsibilities** **(N = 4)**	“it’s been ok over that period but it has been full focus on that, it hasn’t been like I’ve been able to work during that time, those days, or study” [PP-03] “Since she needs special support a certain competence is needed that you may not have as a parent, I think” [PP-09-10]	* b140 Attention functions d760 Family relationships * d820 School education * e125 Products and technology for communication * e130 Products and technology for education e310 Immediate family e330 People in positions of authority e360 Other professionals
		**Remind and deal with new routines** **(N = 15)**	“But now, considering that they have to be outside I have to force him [to change clothes] motivate him for it and have that struggle with him. And it has been really difficult for him to adjust” [PP-02] “But it’s a bit like you have that pressure that you have a child out there after all that needs support and social contact. When you will, I don’t really have time to for that if I do what I should for my workday…She has selective eating as well. It’s like you sit and then, if I go out to prepare lunch I can’t only make lunch for myself. Then I have to prepare something for her as well” [PP-09-10] “Right now I sit for an hour. But then usually my kids get bored and then it’s a struggle anyway. But I just try to manage. But it’s hard. I’ve been on sick leave half time for a period […] I have an anxiety disorder that has become much harder because of COVID” [PP-67]	* b152 Emotional functions * b164 Higher level cognitive functions * d230 Carrying out daily routine * d630 Preparing meals * e310 Immediate family
	REDUCED STRESSORS	**Fewer everyday routines** **(N = 5)**	“Kind of, that it’s a, the mornings used to consist of 20 activities and now they consist of three. It’s amazing. I hope that may stick around even after COVID” [PP-67] “Also the ability, you have gotten much more strength and energy to do things with the family since you don’t commute but then you might rather do fun things together instead” [PP-75]	* b130 Energy and drive functions b164 Higher level cognitive functions d230 Carrying out daily routine d760 Family relationships * d820 School education d850 Remunerative employment e310 Immediate family
**THOUGHTS ABOUT THE FUTURE**	ANXIETY	**Worried about the future of my society** **(N = 4)**	“it’s an enormous economic hit to society. And we already see the tendencies how you save a lot particularly on people with neuropsychiatric disorders unfortunately” [PP-13]	e570 Social security services, systems, and policies e580 Health services, systems, and policies ** e565 Economic services, systems and policies *** e545 Civil protection services, systems and policies
		**Worried about my future** **(N = 24)**	“I worry a bit about if I will feel worse and if it will be a bigger shock to return to school” [YP-08] “So that’s a thing that I think about that I worry a bit about going back to how it was before. That I will take on too much. Then not have any energy left” [YP-78] “That my spot or whatever you want to call it, will be outrivalled by patients with COVID-19 that have PTSD” [AP-12] “And it’s like, if the world economy does not work then my economy won’t work that well either. So that’s probably what I’m most worried about” [AP-33] “It gets harder to meet new people and participate in social activities when I’ve stayed away a longer period of time” [AP-14] “on the way to adulthood, that is on the boundary now between finishing her studies in a few years and go into adulthood. Where it’s been clear that it’s partly delayed because of COVID-19 when she has to re-do a year” [PP-09-10]	* b130 Energy and drive functions b152 Emotional functions b160 Thought functions * d230 Carrying out daily routine d240 Handling stress and other psychological demands d910 Community life e570 Social security services, systems, and policies e580 Health services, systems, and policies ** e565 Economic services, systems and policies
	OPTIMISM	**Lessons in society in general** **(N = 15)**	“That people learn how to deal with this in the future if it happens again. That you kind of can have things at home in advance, so you don’t start hoarding when everyone really needs it. And how you take care of yourself and others” [YP-11] “After Corona…Well, to be able to hug again, I hope” [AP-01]	* d750 Informal social relationships d850 Remunerative employment d920 Recreation and leisure e580 Health services, systems, and policies *** e545 Civil protection services, systems and policies ** e298 Natural environment and human-made changes to environment, other specified e315 Extended family e320 Friends
		**Digital options** **(N = 5)**	“maybe you kind of want to keep video-calls later too when it gets more normal” [AP-04] “So I think that’s been pretty nice, that I have proved that this works. And that you hopefully can keep some of these elements when we go back to some kind of normal again. That you maybe can have that a few times, that this I can do digitally” [AP-06] “And I actually think that this is something that all of society has learnt. I think that should be utilized. Both like in school, teachers has gotten a crash course in digital education instead” [PP-07]	* e125 Products and technology for communication * d360 Using communication devices and techniques d850 Remunerative employment d920 Recreation and leisure
		**Developed school platforms** **(N = 7)**	“The only thing I guess I am happy about is as a teacher that it has been highlighted how important school and education is in different ways. I guess that’s it, as I said, as I mentioned previously what function school fills that have different functions” [PP-03] “if it happens again or if this gets drawn out. You wish that schools got more support. I think there are many lessons where you could make this easier for all involved.” [PP-71]	* e125 Products and technology for communication * d360 Using communication devices and techniques * d820 School education e360 Other professionals
		**Lessons about myself** **(N = 2)**	“But I realise too how stressed I become from demands when I can’t really del with them. That’s been a pretty good insight” [AP-12]	* d240 Handling stress and other psychological demands * d220 Undertaking multiple tasks
**DAILY LIFE AT HOME**	INCREASING PRESSURE	**When you lose outside structure days become more challenging/stressful** **(N = 11)**	“it’s not really that voluntary to not participate in activities because for me it has felt like the advantages of maintaining routines would have been bigger” [PP-02] “You made up little chores. Like household- or garden work that he had to solve. That way he kept busy” [PP-13] “But structure from the outside and structure that I create myself. And when you lose what is outside it becomes a bit harder” [YP-78]	* b134 Sleep functions d160 Focusing attention * d210 Undertaking a single task * d230 Carrying out daily routine * d920 Recreation and leisure * b164 Higher level cognitive functions * b130 Energy and drive functions
		**Strained familial relationships** **(N = 16)**	“You know, it’s wearying, it is. You are on each other all the time. Both have a need for alone time of course. Yeah, there is not much of that. You can go for a walk but then” [AP-12] “A “regular” living situation throughout the spring. Then it was really over the last week we decided to get a divorce” [AP-65] “Then communication gets very limited and he gets very squared. Then I raised my hand and he spinning metal, this medal, metal towards me and my finger broke” [PP-13]	b122 Global psychosocial functions * b125 Dispositions and intra-personal functions * b126 Temperament and personality functions * b152 Emotional functions * b164 Higher level cognitive functions * d710 Basic interpersonal interactions * d720 Complex interpersonal interactions * d760 Family relationships * d770 Intimate relationships
		**Generally dissatisfied with routines** **(N = 4)**	“My sleeping has always been pretty bad and it still is” [AP-12]	* b134 Sleep functions * d630 Preparing meals * d230 Carrying out daily routine
	RELEASE AND SUPPORT	**Release of everyday demands** **(N = 11)**	“First of all, you don’t have to go to school. And more than that you didn’t have to prepare as much, pack your bag and stuff. You could kind of have your lessons in you pyjamas if you wanted to, it sort of didn’t matter. Then I could wake up later. Which was nice” [YP-78] “I think stuff like my sleep that it’s worked better because my schedule has been marginally reduced and I can do it” [AP-06] “Well, since she doesn’t get all that, impressions at school, and have to socialize with lots of group-work and talk to people and sounds everywhere. Then she’s been much more alert, then she has energy to socialize with others” [PP-07]	* d210 Undertaking a single task * d220 Undertaking multiple tasks * d230 Carrying out daily routine * d820 School education d850 Remunerative employment d920 Recreation and leisure
		**Stronger family connections** **(N = 15)**	“Yeah. It’s, really it’s, we have been able to see each other more. Since he [brother] is not with his friends at the moment but he’s here. So it’s pretty cool and fun that he is here” [YP-72] “And it’s been so great to be a part of and contribute to that and feel like I’ve sat and talked to my daughter about all sorts of things” [AP-65] “Within the family it’s actually only been positive for us. It has gotten us more attached than detached. It’s the opposite, we have gotten it much better” [PP-75]	d760 Family relationships
		**Generally satisfied with routines** **(N = 13)**	“Well, I sleep pretty well. Food, hm, is as usual I guess” [YP-72] “So we have pretty established routines. And we largely hold on to them because it usually turns out for the best” [PP-07]	* b130 Energy and drive functions b134 Sleep functions b152 Emotional functions d230 Carrying out daily routine * d570 Looking after one’s health
**THOUGHTS ABOUT COVID-19 RESTRICTIONS**	DISTRESS	**Understanding and implementing restrictions cause distress** **(N = 19)**	“I have to distance to meters even though we live under the same roof. That can be a bit tricky” [AP-01] “But it would never enter my mind to just go and wash my hands, so I never do that. And I really don’t like, you know I don’t like ehm hand sanitizer. So I never use hand sanitizer, sort of how it feels or how it smells, so I never use that” [AP-05] “Everyone can do aerobics but then maybe then you consider that perhaps it’s this particular group that would need it most of all and could continue with the activity” [PP-13] “somebody to hold my hand and say you’re doing it right, this is how you’re supposed to do it or I don’t know. I think it’s difficult. Not be bothered that everyone is doing things differently all the time” [PP-70]	d177 Making decisions * d230 Carrying out daily routine d240 Handling stress and other psychological demands * d570 Looking after one’s health * d920 Recreation and leisure *** e545 Civil protection services, systems and policies e310 Immediate family e320 Friends
		**I get upset when other people don’t comply with restrictions** **(N = 12)**	“Well, you know, it’ s like, people are acting like COVID is finished. I don’t know if it’s because they want to feel a safety that things are normal, if it’s routines, I don’t know. But they don’t take it seriously, don’t keep their distance, don’t cough in their armpit” [YP-72] “He’s been incredible frustrated by the way the school has not followed the Public Health Agency’s guidelines and stuff. It has really affected his every day to a large extent” [PP-71]	* b152 Emotional functions * d570 Looking after one’s health *** e545 Civil protection services, systems and policies
	UNDERSTANDING	**More aggressive measures would have had a worse effect** **(N = 12)**	“I think this has worked better for me. I’ve tried to imagine what it would be like to have a lockdown 60 days you can’t go outside the door method” [AP-65] “No I think it’s been great to have a choice. I don’t think we would have been able to manage that kind of total limitation there has been in other places” [PP-02] “for us this strategy has worked fine that we get to take own responsibility because we have taken responsibility as much as we have been able to. The of course if we would have isolated more it would have affected us even more” [PP-74]	e580 Health services, systems, and policies *** e545 Civil protection services, systems and policies
		**Already observe physical distancing** **(N = 10)**	“I have been on sick leave over soon to be two years, for depression. And then I was already, or because of that and already before I have been more or less self-isolated” [AP-35] “We don’t have kids with many recreational activities and we don’t kind of socialize every weekend with other families with kids and we don’t travel and run around like that. Where there are actually restrictions now” [PP-09-10]	* d720 Complex interpersonal interactions d730 Relating with strangers ** d729 General interpersonal interactions, other specified and unspecified
		**Acceptance of restrictions** **(N = 24)**	“Well it’s not positive but you still understood why and that it’s still, well you know, well hopefully that you know its for a limited period kind of like” [AP-04] “So I think that would it continue in the same way through the fall, we would probably be able to handle that so” [PP-03] “But he hasn’t thought it’s been difficult or… he hasn’t expressed any worry regarding Corona or anything like that” [PP-66]	* d230 Carrying out daily routine d240 Handling stress and other psychological demands d250 Managing one’s own behavior * d570 Looking after one’s health *** e545 Civil protection services, systems and policies

N—number of individuals with codes with each separate code. YP—young person; AP—adult person; PP —parent person. […]—clarified by coder; shortened quote; pseudonymized information in the interview transcript. * ICF core set re-coded 3rd or 4th level code. ** ICF not in core sets 2nd level code. *** ICF not in core sets re-coded 3rd or 4th level code.

**Table 4 ijerph-19-01268-t004:** Themes, codes, and illustrative quotes following an interview with a focus group of representatives of interest organizations.

Themes	Codes	Quotes
IMPLEMENTATION IS CHALLENGING DESPITE CLEAR GUIDLINES	**Adaptation of guidelines in everyday life** **(N = 4)**	Cause I have a runny nose but it disappears when I take my pollen allergy medication or whatever it may be. So it is difficult when there are clear guidelines but you are literal [R5] Regarding that I have experienced that there is a great worry and also if everything is clear now you want to, for how long. [R5] There may also be autistic people, older that still lives with their parents and are a bit, have no idea what they should do now. If they should also remain completely indoors. [R1] Occurs that people, as it were, have started washing their hands. Following this pandemic. [R4]
	**Worried about how others implement guidelines** **(N = 3)**	…our group needs to be separated. We have agreed to follow these rules. But there are others there and then we can’t stay. [R4] Because I know that members call and ask when we are going to be outdoors, that all will keep their distance. Well, that’s hard for us to guarantee. [R3] There is less risk when to attend activities with us because we also keep our distance. [R4]
	**Satisfied with information about the guidelines** **(N = 2)**	I think that the Public Health Agency’s guidelines that have been available, on their website, they have had one of those checklists you can follow very clearly and great. [R5] There have been support in various forms I think. Like the habilitation services and [healthcare services] region have developed if you have an intellectual disability. [R3] And I think there are several of our organizations that have helped spreading this information surrounding COVID and how to deal with it and what guidelines exists. [R3]
	**Adapting associations activities for members** **(n = 4)**	…it’s not like they ask me how to interpret COVID-19 restrictions or the recommendations. If we are having an activity with our members have to try to negotiate. What do you require to dare to attend a member activity. What do you require. Is it ok if we do it like this. Yes, ok. Then we do it that way. [R4] Well, it’s like what I told you locally there are physical meetings outdoors. And it works fine. [R1] we had to cancel a meeting or two because the leaders were sick. Other than that we haven’t cancelled anything we’ve had planned. [R5] But then it’s also that we have cancelled some activity-groups because partly leaders aren’t comfortable meeting digitally. And physically they aren’t willing to do it because of the risk of infection. [R3] And some others have been like upset that we have changed our activities because they think it is exaggerated. You should just have it like we usually have it. [R3]
WHEN SUPPORT STRUCTURES DIMINISH, DEMANDS INCREASE	**Sudden loss of everyday support** **(N = 3)**	The daily activity centers closed in Stockholm and it was very sudden and without any real preparations. [R2] We’ve had contact with members that say just that, the living support service have stopped showing up. Or that they themselves have renounced the living support to not get infected. [R3] That means that some of our members did not find out until the Monday when they were going to their, if you, those I’ve spoken to have had a job-oriented daily activity and been out to important, well, jobs. So when they got there they weren’t allowed to enter. That creates worry. [R3] The staff that had been in the communal daily activity centers they suddenly also got, what do you say, re-directed to work in assisted living because they were more needed there. That meant that the people that lives there suddenly new staff came there. Which also created more stress and worry, because who is that person coming here. [R3]
	**Loss of everyday structure** **(N = 3)**	I have agreed with one member that have renounced their living support service that helps her clean that I send a text-message to her that I know, have you cleaned now. [R4] So there are a lot of things that changes and if you are very dependent on your structure in your everyday life everything might asily fall then. [R5] I think it came a bit quick for some sometimes. But they have probably found strategies after a while. But those have strategies they have found themselves. [R3] It isn’t always easy to figure out what to do instead. In my experience many of our members feel psychologically worse. [R5]
	**Parental responsibilities increase** **(N = 3)**	In higher education institutions they have closed the schools. And to be home can be very exhausting so to speak for the whole family. [R2] Then for the younger children that have been in school it has been in reality a relief that that routine has continued. [R2] Many parents that, parents of children with NDD [neurodevelopmental disabilities] has gotten it really tough. They get no relief now. […] So the parents cannot work at the same time as you have children at home. Even if it is children at high school level for instance. [R5]
INCREASED SENSE OF INCLUSION IN SOCIETY	**Remote alternatives increase participation** **(N = 4)**	Specifically students that have pro-longed absenteeism or with a high degree of school-absence or where it simply hasn’t worked in school have heard from many that suddenly the person got approved school-presence because a system has been created where it works. [R5] So it’s, I know that some have had a check-in with the teacher at certain times throughout the day. So they study on their own and then you have a check-in and stuff. But they also have had contact, digitally, teachers are more accessible in that they can write in a chat or e-mail or in some way. As it isn’t normally. Then those students that aren’t going to school they can now suddenly have contact with the teacher. [R5] Now you can with remote schooling you got approved school-presence in different ways. By for instance login at certain time-points and stuff. After that I know several that have gotten more motivated to complete tasks because then there is still a chance to pass. [R5] I spoke with a pretty upset mum in the beginning of COVID-19 and corona that were pretty angry because they had requested these things for so long and all of a sudden in just like a week it works to do it. [R3] To transport to work, go by subway, go by train. That might steal the energy of half a working day in itself. So when you don’t have to do that you notice that god no I can, now I can really work. [R5] And in the association then. We’re a very social association so in the district and in the interest organization many are friends. And then we can, then it’s like, in a way easier to meet actually. [R4] I can only say that for instance within the [healthcare] region nowadays they often have telephone- and video-meetings. Which also have meant that more have attended their doctors’ appointments. [R3]
	**Sense of normalization of lived experiences** **(N = 2)**	That you feel like you don’t have to meet people you don’t want to meet people you otherwise have to meet. Stuff like that. [R4] So now that everyone has to live like autistic people in some respect, we’re supposed to isolate, we’re supposed to be alone. Like, then can at least some aspects become better for us. [R4] That you, if you work from home and that doesn’t sound weird in any way but it’s, well, it’s because of COVID. [R4] That it’s ok to keep your distance and you don’t have to have the forced socialization. [R3]
	**Digitalization of associations organization** **(N = 4)**	While before it would only be a hassle, ok I need to sit by a computer, how does it work and stuff like that. It happens that people are appreciating things in a different way. [R4] I think this will actually permanently change our way of working. It is a pretty significant part and I see that as only beneficial. [R2] Like, physical lectures during the spring. And we almost cancelled them and then when we got this digital tool we had the opportunity to have it that way instead. There were members that thought it was absolutely amazing. [R3]
	**Optimism for the future** **(N = 4)**	Perhaps it might not be possible to offer in all schools but maybe you might find schools that can be digital and that can become schools for people that is suited for. I think that type of specialization you should be able to support and consider from the side of the municipalities. [R1] We are now also discussing contact with healthcare providers that we might have increased competence for our disabled, NDD, from telehealth doctors that can specialize. [R2] Then I’m also thinking about the digital we discussed previously that it has actually entered and developed in society and increase possibilities rather than exclusions. [R3] …to deal with this kinds of situations will be better in the future. [R4] The only silver lining we’ve touched upon that perhaps with a change in how we work and digital distance-work might create possibilities for some. [R2]
LOSS OF IMPORTANT RELATIONSHIPS AND PROSPECTS FOR FUTURE	**Sense of social loss** **(N = 4)**	Where some have been like, I won’t be seeing anybody at all now, I isolate completely. Because you interpret it all quite literally. [R3] And I think it is pretty significant that some autistic people have been very lonely during all this. It might have been assisted living support, the only person they have seen all week. Got quite a few panic-calls that has been about that. [R1] …cannot see their parents that may help a lot in their lives. Or the safety to be able to see them. So we hear a lot of the negative aspects. That they create even more worry to be alone and not be able to see others physically. Or to be able to attend the physical activities the association has. [R5]
	**Worried about inclusion of autism community in the future** **(N = 2)**	…many of our members have had difficulties finding jobs and now many are, unemployment has increased. [R2] And I’m afraid that some of what is the welfare and that our members enjoy and create, it helps create their lives, can be threatened. [R2] So there is actually some concern within the world of associations because of the financial contributions. Will we get fewer contributions next year. [R3]

N—number of coded responders per code. R—abbreviation of responder used as psudonymised identifier of each quote.

## Data Availability

The data presented in this study are available upon request from the corresponding author.

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
