# Peer review of "Effects of the Early Phase of COVID-19 on the Autistic Community in Sweden: A Qualitative Multi-Informant Study Linking to ICF"

_ijerph, 2022, doi:10.3390/ijerph19031268_

Round 1

Reviewer 1 Report

The paper addresses an important topic. COVID has had a disproportionate impact on people with disabilities -- not just in morbidity but in the social and economic impact. A fast growing literature is making that case. And I have seen a number of articles on the impact of COVID on people with mental health issues -- but I have not seen a similar emphasis on autistic people. So this is a welcome addition to the literature.  

I have two points that basically just deal with presentation -- not with the methods, analysis, or findings. A minor one, and one that will require more work.

1) Autism encompasses a very broad spectrum of social and communicative issues. The subjects of this study are clearly on the higher end of functioning and so that should be made clear. The very fact that they are being interviewed and can express themselves this well demonstrates this. This is not a criticism, but merely a statement that that should be made clear.

2) What is special about the impact (in type or degree) of COVID on autistic people must be made clearer. The question is WHY do the researchers choose to look at autistic people? Presumable because they have certain communication and social issues -- things like difficulty with transitions, need for routine, different social needs and abilities, etc. Therefore, the interaction between that and the COVID experience needs to be highlighted better. Neurologically typical people also felt lonely, also had disruptions to their routines, etc.  What is special about autistic people that made their experiences different -- in type or degree?  In truth, that is in the paper already but the way it is organized I had to tease it out myself -- a sentence here a paragraph there. Sometimes it is explicit -- like when talking about the positive impacts of working virtually -- and sometimes I can read it between the lines.  I would modify the paper in a way that lays out first the strengths and challenges that autistic people have/face -- and then show how that interacts with the limitations and disruptions created by COVID.  It's all there in the paper, but I think it would be better if it were brought out more.

Reviewer 2 Report

In this study, Fridell et al.; interviewed autistic children, autistic adults, parents of autistic children representatives of autism interest groups. The interviews were conducted in the early stage of the covid-19 pandemic.
The aim of the study was to explore how the covid pandemic has affected the autism community – the groups mentioned above. 

The study has its merits as well as its limitations. The field of study is relevant as it could be suspected that people with autism would be more affected by the pandemic and the imposed restrictions. One particularly interesting aspect that the authors ought to discuss is the differences in the imposed restrictions between different countries and what this might mean for the autistic community.

The paper is quite long but well written. The authors might want to consider moving some of the results into supplementary material (e.g. the specific answers from the participants). The conclusions are not surprising but still constitute a valuable addition to the literature. Overall, the paper is suitable for publication in IJERPH, though I recommend that the authors revise to alleviate the following concerns

Concern 1 – Lumping together heterogeneous groups.
In total, the authors performed and analyzed 38 interviews – a respectable number when it comes to a qualitative investigation. Nevertheless, the subgroups included under the autism community umbrella term "autism community" are small and quite diverse. It seems unlikely that the imposed restrictions would affect children with autism and representatives of autism interest groups in the same way. It is therefore questionable whether one can lump these groups together in the way that the authors have done in the results and in the rest of the paper. The authors should describe whether they saw any differences between the different groups, or at the very least, describe this issue to the readers. 

Concern 2 – No control group
There is no control group, so based on the data here, it is impossible to tell if the effects on the autistic community differ from the effects on other people. It seems likely that many of the patterns they observed here would also be found in a control group – had it been included. 

Concern 3 – Unclear age span
There is conflicting information about the age distribution of the children in the study? How many below 14 years were there among the participants? This is also relevant since the authors state that children 12-14 were given a special set of questions (Row 212-214). How many children were given this set of questions?

Row 167: "7 autistic children aged 12-17"
Table 1: Range for young people 14-17 (?)

Minor comments
Row 12-13: Should be [of] the pandemic 
Row 21: "The study interviewed" 
Row 231: Md = mean?
Row 242: "Ranged 8844 words" – not a range
Row 252: "on member"
Row 265: The authors should state somewhere whether the two different raters agreed (maybe I missed it?)
Row 748-751: Sentence sounded strange to me...
